# Factor Group-Sparse Regularization for Efficient Low-Rank Matrix Recovery

**Jicong Fan**
Cornell University
Ithaca, NY 14850
jf577@cornell.edu

**Lijun Ding**
Cornell University
Ithaca, NY 14850
ld446@cornell.edu

**Yudong Chen**
Cornell University
Ithaca, NY 14850
yudong.chen@cornell.edu

**Madeleine Udell**
Cornell University
Ithaca, NY 14850
udell@cornell.edu

## Abstract

This paper develops a new class of nonconvex regularizers for low-rank matrix recovery. Many regularizers are motivated as convex relaxations of the *matrix rank* function. Our new factor group-sparse regularizers are motivated as a relaxation of the *number of nonzero columns* in a factorization of the matrix. These nonconvex regularizers are sharper than the nuclear norm; indeed, we show they are related to Schatten-$p$ norms with arbitrarily small $0 < p \leq 1$. Moreover, these factor group-sparse regularizers can be written in a factored form that enables efficient and effective nonconvex optimization; notably, the method does not use singular value decomposition. We provide generalization error bounds for low-rank matrix completion which show improved upper bounds for Schatten-$p$ norm reglarization as $p$ decreases. Compared to the max norm and the factored formulation of the nuclear norm, factor group-sparse regularizers are more efficient, accurate, and robust to the initial guess of rank. Experiments show promising performance of factor group-sparse regularization for low-rank matrix completion and robust principal component analysis.

## 1 Introduction

Low-rank matrices appear throughout the sciences and engineering, in fields as diverse as computer science, biology, and economics [1]. One canonical low-rank matrix recovery problem is low-rank matrix completion (LRMC) [2, 3, 4, 5, 6, 7, 8, 9, 10], which aims to recover a low-rank matrix from a few entries. LRMC has been used to impute missing data, make recommendations, discover latent structure, perform image inpainting, and classification [11, 12, 1]. Another important low-rank recovery problem is robust principal components analysis (RPCA) [13, 14, 15, 16, 17], which aims to recover a low-rank matrix from sparse but arbitrary corruptions. RPCA is often used for denoising and image/video processing [18].

**LRMC**    Take LRMC as an example. Suppose $\boldsymbol{M} \in \mathbb{R}^{m \times n}$ is a low-rank matrix with rank$(\boldsymbol{M}) = r \ll \min(m, n)$. We wish to recover $\boldsymbol{M}$ from a few observed entries. Let $\Omega \subset [m] \times [n]$ index the observed entries. Suppose card$(\Omega)$, the number of observations, is sufficiently large. A natural idea is to recover the missing entries by solving

$$\underset{\boldsymbol{X}}{\text{minimize}} \ \text{rank}(\boldsymbol{X}), \ \text{subject to} \ \mathcal{P}_\Omega(\boldsymbol{X}) = \mathcal{P}_\Omega(\boldsymbol{M}), \tag{1}$$

where the operator $\mathcal{P}_\Omega : \mathbb{R}^{m \times n} \to \mathbb{R}^{m \times n}$ acts on any $\boldsymbol{X} \in \mathbb{R}^{m \times n}$ in the following way: $(\mathcal{P}_\Omega(\boldsymbol{X}))_{ij} = \boldsymbol{X}_{ij}$ if $(i, j) \in \Omega$ and $0$ if $(i, j) \notin \Omega$. However, since the direct rank minimization problem (1) is NP-hard, a standard approach is to replace the rank with a tractable surrogae $R(\boldsymbol{X})$ and solve

$$\underset{\boldsymbol{X}}{\text{minimize}} \ R(\boldsymbol{X}), \text{ subject to } \mathcal{P}_\Omega(\boldsymbol{X}) = \mathcal{P}_\Omega(\boldsymbol{M}). \tag{2}$$

Below we review typical choices of $R(\boldsymbol{X})$ to provide context for our factor group-sparse regularizers.

**Nuclear and Schatten-$p$ norms**   One popular convex surrogate function for the rank function is the nuclear norm (also called trace norm), which is defined as the sum of singular values:

$$\|\boldsymbol{X}\|_* := \sum_{i=1}^{\min(m,n)} \sigma_i(\boldsymbol{X}), \tag{3}$$

where $\sigma_i(\boldsymbol{X})$ denotes the $i$-th largest singular value of $\boldsymbol{X} \in \mathbb{R}^{m \times n}$. Variants of the nuclear norm, including the *truncated nuclear norm* [19] and *weighted nuclear norm* [20], sometimes perform better empirically on imputation tasks.

The Schatten-$p$ norms[1] with $0 \le p \le 1$ [21, 22, 23] form another important class of rank surrogates:

$$\|\boldsymbol{X}\|_{\mathrm{S}p} := \Big( \sum_{i=1}^{\min(m,n)} \sigma_i^p(\boldsymbol{X}) \Big)^{1/p}. \tag{4}$$

For $p = 1$, we have $\|\boldsymbol{X}\|_{\mathrm{S}_1}^1 = \|\boldsymbol{X}\|_*$, the nuclear norm. For $0 < p < 1$, $\|\boldsymbol{X}\|_{\mathrm{S}_p}^p$ is a nonconvex surrogate for $\mathrm{rank}(\boldsymbol{X})$. In the extreme case $p = 0$, $\|\boldsymbol{X}\|_{\mathrm{S}_0}^0 = \mathrm{rank}(\boldsymbol{X})$, which is exactly what we wish to minimize. Thus we see $\|\boldsymbol{X}\|_{\mathrm{S}_p}^p$ with $0 < p < 1$ interpolates between the rank function and the nuclear norm. Instead of (1), we hope to solve

$$\underset{\boldsymbol{X}}{\text{minimize}} \ \|\boldsymbol{X}\|_{\mathrm{S}_p}^p, \text{ subject to } \mathcal{P}_\Omega(\boldsymbol{X}) = \mathcal{P}_\Omega(\boldsymbol{M}), \tag{5}$$

with $0 < p \le 1$. Smaller values of $p$ ($0 < p \le 1$) are better approximations of the rank function and may lead to better recovery performance for LRMC and RPCA. However, for $0 < p < 1$ the problem (5) is nonconvex, and it is not generally possible to guarantee we find a global optimal solution. Even worse, common algorithms for minimizing the nuclear norm and Schatten-$p$ norm cannot scale to large matrices because they compute the singular value decomposition (SVD) in every iteration of the optimization [2, 3, 24].

**Factor regularizations**   A few SVD-free methods have been develoepd to recover large low-rank matrices. For example, the work in [25, 26] uses the well-known fact that

$$\|\boldsymbol{X}\|_* = \min_{\boldsymbol{AB}=\boldsymbol{X}} \|\boldsymbol{A}\|_F \|\boldsymbol{B}\|_F = \min_{\boldsymbol{AB}=\boldsymbol{X}} \frac{1}{2}\big(\|\boldsymbol{A}\|_F^2 + \|\boldsymbol{B}\|_F^2\big), \tag{6}$$

where $\boldsymbol{A} \in \mathbb{R}^{m \times d}$, $\boldsymbol{B} \in \mathbb{R}^{d \times n}$, and $d \ge \mathrm{rank}(\boldsymbol{X})$. For LRMC they suggest solving

$$\underset{\boldsymbol{A}, \boldsymbol{B}}{\text{minimize}} \ \frac{1}{2}\|\mathcal{P}_\Omega(\boldsymbol{M} - \boldsymbol{AB})\|_F^2 + \frac{\lambda}{2}\big(\|\boldsymbol{A}\|_F^2 + \|\boldsymbol{B}\|_F^2\big). \tag{7}$$

In this paper, we use the name *factored nuclear norm* (F-nuclear norm for short) for the variational characterization of nuclear norm as $\min_{\boldsymbol{AB}=\boldsymbol{X}} \frac{1}{2}\big(\|\boldsymbol{A}\|_F^2 + \|\boldsymbol{B}\|_F^2\big)$ in (6). This expression matches the nuclear norm when $d$ is chosen large enough. Srebro and Salakhutdinov [27] proposed a weighted F-nuclear norm; the corresponding formulation of matrix completion is similar to (7). Note that to solve (7) we must first choose the value of $d$. We require $d \ge r := \mathrm{rank}(\boldsymbol{M})$ to be able to recover (or even represent) $\boldsymbol{M}$. Any $d \ge r$ gives the same solution $\boldsymbol{AB}$ to (7). However, as $d$ increases from $r$, the difficulty of optimizing the objective increases. Indeed, we observe in our experiments that the recovery error is larger for large $d$ using standard algorithms, particularly when the proportion of observed entries is low. In practice, it is difficult to guess $r$, and generally a very large $d$ is required. The methods of [28] and [29] estimate $r$ dynamically.

Another SVD-free surrogate of rank is the max norm, proposed by Srebro and Shraibman [30]:

$$\|\boldsymbol{X}\|_{\max} = \min_{\boldsymbol{AB}=\boldsymbol{X}} \big( \max_i \|\boldsymbol{a}_i\| \big)\big( \max_j \|\boldsymbol{b}_j\| \big), \tag{8}$$

where $\boldsymbol{a}_i$ and $\boldsymbol{b}_j$ denotes the $i$-th row of $\boldsymbol{A}$ and the $j$-th row of $\boldsymbol{B}^T$ respectively. Lee et al. [31] proposed several efficient algorithms to solve optimization problems with the max norm. Foygel and Srebro [5] provided recovery guarantees for LRMC using the max norm as a regularizer.

Another very different approach uses implicit regularization. Gunasekar et al. [32] show that for full dimensional factorization without any regularization, gradient descent with small enough step size and initialized close enough to the origin converges to the minimum nuclear norm solution. However, convergence slows as the initial point and step size converge to zero, making this method impractical.

Shang et al. [33] provided the following characterization of the Schatten-1/2 norm:

$$\|\boldsymbol{X}\|_{\mathrm{S}_{1/2}} = \min_{\boldsymbol{AB}=\boldsymbol{X}} \|\boldsymbol{A}\|_* \|\boldsymbol{B}\|_* = \min_{\boldsymbol{AB}=\boldsymbol{X}} \big( \tfrac{\|\boldsymbol{A}\|_* + \|\boldsymbol{B}\|_*}{2} \big)^2. \tag{9}$$

Hence instead of directly minimizing $\|\boldsymbol{X}\|_{\mathrm{S}_{1/2}}^{1/2}$, one can minimize $\|\boldsymbol{A}\|_* + \|\boldsymbol{B}\|_*$, which is much easier when $r \leq d \ll \min(m,n)$. But again, this method and its extension $\|\boldsymbol{A}\|_* + \frac{1}{2}\|\boldsymbol{B}\|_F^2$ proposed in [34] require $d \geq r$, and the computational cost increases with larger $d$. Figure 1(d) shows these approaches are nearly as expensive as directly minimizing $\|\boldsymbol{X}\|_{\mathrm{S}_p}^p$ when $d$ is large. We call the regularizers $\min_{\boldsymbol{AB}=\boldsymbol{X}}(\|\boldsymbol{A}\|_* + \|\boldsymbol{B}\|_*)$ and $\min_{\boldsymbol{AB}=\boldsymbol{X}}(\|\boldsymbol{A}\|_* + \frac{1}{2}\|\boldsymbol{B}\|_F^2)$ the *Bi-nuclear norm* and $F^2$+*nuclear norm* respectively.

**Our methods and contributions**   In this paper, we propose a new class of factor group-sparse regularizers (FGSR) as a surrogate for the rank of $\boldsymbol{X}$. To derive our regularizers, we introduce the factorization $\boldsymbol{AB} = \boldsymbol{X}$ and seek to minimize the number of nonzero columns of $\boldsymbol{A}$ or $\boldsymbol{B}^T$. Each factor group-sparse regularizer is formed by taking the convex relaxation of the number of nonzero columns. These regularizers are convex functions of the factors $\boldsymbol{A}$ and $\boldsymbol{B}$ but capture the nonconvex Schatten-$p$ (quasi-)norms of $\boldsymbol{X}$ using the nonconvex factorization constraint $\boldsymbol{X} = \boldsymbol{AB}$.

- We show that these regularizers match arbitrarily sharp Schatten-$p$ norms: for each $0 < p' \leq 1$, there is some $p < p'$ for which we exhibit a factor group-sparse regularizer equal to the sum of the $p$th powers of the singular values of $\boldsymbol{X}$.
- For a class of $p$, we propose a generalized factorization model that enables us to minimize $\|\boldsymbol{X}\|_{\mathrm{S}_p}^p$ without performing the SVD.
- We show in experiments that the resulting algorithms improve on state-of-the-art methods for LRMC and RPCA.
- We prove generalization error bounds for LRMC with Schatten-$p$ norm regularization, which explain the superiority of our methods over nuclear norm minimization.

**Notation**   Throughout this paper, $\|\cdot\|$ denotes the Euclidean norm of a vector argument. We factor $\boldsymbol{X} \in \mathbb{R}^{m \times n}$ as $\boldsymbol{A} = [\boldsymbol{a}_1, \boldsymbol{a}_2, \cdots, \boldsymbol{a}_d] \in \mathbb{R}^{m \times d}$ and $\boldsymbol{B} = [\boldsymbol{b}_1, \boldsymbol{b}_2, \cdots, \boldsymbol{b}_d]^T \in \mathbb{R}^{d \times n}$, where $d \geq r := \mathrm{rank}(\boldsymbol{X})$, and $\boldsymbol{a}_j$ and $\boldsymbol{b}_j$ are column vectors. Without loss of generality, we assume $m \leq n$. All proofs appear in the supplement.

## 2   FGSRs match Schatten-$p$ norms with $p = \frac{2}{3}$ or $\frac{1}{2}$.

Let $\mathrm{nnzc}(\boldsymbol{A})$ denote the number of nonzero columns of matrix $\boldsymbol{A}$. Write the rank of $\boldsymbol{X} \in \mathbb{R}^{m \times n}$ as

$$\mathrm{rank}(\boldsymbol{X}) = \min_{\boldsymbol{AB}=\boldsymbol{X}} \mathrm{nnzc}(\boldsymbol{A}) = \min_{\boldsymbol{AB}=\boldsymbol{X}} \mathrm{nnzc}(\boldsymbol{B}^T) = \min_{\boldsymbol{AB}=\boldsymbol{X}} \frac{1}{2}\big( \mathrm{nnzc}(\boldsymbol{A}) + \mathrm{nnzc}(\boldsymbol{B}^T) \big). \tag{10}$$

Now relax: notice $\mathrm{nnzc}(\boldsymbol{A}) \geq \sum_{j=1}^d \|\boldsymbol{a}_j\|$ when $\|\boldsymbol{a}_j\| \leq 1$ for each column $j$. We show that using this relaxation in (10) gives a factored characterization of the Schatten-$p$ norm with $p = \frac{1}{2}$ or $\frac{2}{3}$.

**Theorem 1.** *Fix* $\alpha > 0$. *For any matrix* $\boldsymbol{X} \in \mathbb{R}^{m \times n}$ *with* $\mathrm{rank}(\boldsymbol{X}) = r \le d \le \min(m, n)$,

$$\min_{\boldsymbol{X} = \sum_{j=1}^{d} \boldsymbol{a}_j \boldsymbol{b}_j^T} \sum_{j=1}^{d} \|\boldsymbol{a}_j\| + \|\boldsymbol{b}_j\| = 2 \sum_{j=1}^{r} \sigma_j^{1/2}(\boldsymbol{X}) \tag{11}$$

$$\min_{\boldsymbol{X} = \sum_{j=1}^{d} \boldsymbol{a}_j \boldsymbol{b}_j^T} \sum_{j=1}^{d} \|\boldsymbol{a}_j\| + \frac{\alpha}{2} \|\boldsymbol{b}_j\|^2 = \frac{3\alpha^{1/3}}{2} \sum_{j=1}^{r} \sigma_j^{2/3}(\boldsymbol{X}). \tag{12}$$

Denote the SVD of $\boldsymbol{X}$ as $\boldsymbol{X} = \boldsymbol{U}_X \boldsymbol{S}_X \boldsymbol{V}_X^T$. Equality holds in equation (11) when $\boldsymbol{A} = \boldsymbol{U}_X \boldsymbol{S}_X^{1/2}$ and $\boldsymbol{B} = \boldsymbol{S}_X^{1/2} \boldsymbol{V}_X^T$; in equation (12), when $\boldsymbol{A} = \alpha^{1/3} \boldsymbol{U}_X \boldsymbol{S}_X^{2/3}$ and $\boldsymbol{B} = \alpha^{-1/3} \boldsymbol{S}_X^{1/3} \boldsymbol{V}_X^T$.

Motivated by this theorem, we define the following factor group-sparse regularizers (FGSR):

$$\mathrm{FGSR}_{1/2}(\boldsymbol{X}) := \frac{1}{2} \min_{\boldsymbol{A}\boldsymbol{B} = \boldsymbol{X}} \|\boldsymbol{A}\|_{2,1} + \|\boldsymbol{B}^T\|_{2,1}. \tag{13}$$

$$\mathrm{FGSR}_{2/3}(\boldsymbol{X}) := \frac{2}{3\alpha^{1/3}} \min_{\boldsymbol{A}\boldsymbol{B} = \boldsymbol{X}} \|\boldsymbol{A}\|_{2,1} + \frac{\alpha}{2} \|\boldsymbol{B}\|_F^2, \tag{14}$$

where $\|\boldsymbol{A}\|_{2,1} := \sum_{j=1}^{d} \|\boldsymbol{a}_j\|$. Theorem 1 shows that $\mathrm{FGSR}_{2/3}$ has the same value regardless of the choice of $\alpha$, which justifies the definition. As a corollary of Theorem 1, we see

$$\mathrm{FGSR}_{1/2}(\boldsymbol{X}) = \sum_{j=1}^{r} \sigma_j^{1/2}(\boldsymbol{X}) = \|\boldsymbol{X}\|_{\mathrm{S}_{1/2}}^{1/2}, \qquad \mathrm{FGSR}_{2/3}(\boldsymbol{X}) = \sum_{j=1}^{r} \sigma_j^{2/3}(\boldsymbol{X}) = \|\boldsymbol{X}\|_{\mathrm{S}_{2/3}}^{2/3}.$$

To solve optimization problems involving these surrogates for the rank, we can use the definition of the FGSR and optimize over the factors $\boldsymbol{A}$ and $\boldsymbol{B}$. It is easier to minimize $\mathrm{FGSR}_{2/3}(\boldsymbol{X})$ than to minimize $\mathrm{FGSR}_{1/2}(\boldsymbol{X})$ because the latter has two nonsmooth terms.

As surrogates for the rank function, $\mathrm{FGSR}_{2/3}$ and $\mathrm{FGSR}_{1/2}$ have the following advantages:

- **Tighter rank approximation.** Compared to the nuclear norm, the spectral quantities in Theorem 1 are tighter approximations to the rank of $\boldsymbol{X}$.
- **Robust to rank initialization.** The iterative algorithms we propose in Sections 4 and 6 to minimize $\mathrm{FGSR}_{2/3}$ and $\mathrm{FGSR}_{1/2}$ quickly force some of the columns of $\boldsymbol{A}$ and $\boldsymbol{B}^T$ to zero, where they remain. Hence the number of nonzero columns is reduced dynamically, and converges to $r$ quickly in experiments: these methods are *rank-revealing*. In constrast, iterative methods to minimize the F-nuclear norm or max norm never produce an exactly-rank-$r$ iterate after a finite number of iterations.
- **Low computational cost.** Most optimization methods for solving problems with the Schatten-$p$ norm perform SVD on $\boldsymbol{X}$ at every iteration, with time complexity of $O(m^2 n)$ (supposing $m \le n$) [21, 22]. In contrast, the natural algorithm to minimize $\mathrm{FGSR}_{2/3}$ and $\mathrm{FGSR}_{1/2}$ does not use the SVD, as the regularizers are simple (not spectral) functions of the factors. The main computational cost is to form $\boldsymbol{A}\boldsymbol{B}$, which has a time complexity of $O(d'mn)$ when the iterates $\boldsymbol{A}$ and $\boldsymbol{B}$ have $d'$ nonzero columns. The complexity of LRMC can be as low as $O(d'\mathrm{card}(\Omega))$.

## 3   Toward exact rank minimization

In the previous section, we developed a factored representation for $\|\boldsymbol{X}\|_{\mathrm{S}_p}^p$ when $p = \frac{2}{3}$ or $\frac{1}{2}$. This section develops a similar representation for $\|\boldsymbol{X}\|_{\mathrm{S}_p}^p$ with arbitrarily small $p$.

**Theorem 2.** *Fix* $\alpha > 0$, *and choose* $q \in \{1, \frac{1}{2}, \frac{1}{4}, \cdots\}$. *For any matrix* $\boldsymbol{X} \in \mathbb{R}^{m \times n}$ *with* $\mathrm{rank}(\boldsymbol{X}) = r \le d \le \min(m, n)$, *we have*

$$\min_{\boldsymbol{X} = \sum_{j=1}^{d} \boldsymbol{a}_j \boldsymbol{b}_j^T} \sum_{j=1}^{d} \frac{1}{q} \|\boldsymbol{a}_j\|^q + \alpha \|\boldsymbol{b}_j\| = (1 + 1/q)\alpha^{q/(q+1)} \sum_{j=1}^{r} \sigma_j^{q/(q+1)}(\boldsymbol{X}), \tag{15}$$

$$\min_{\boldsymbol{X} = \sum_{j=1}^{d} \boldsymbol{a}_j \boldsymbol{b}_j^T} \sum_{j=1}^{d} \frac{1}{q} \|\boldsymbol{a}_j\|^q + \frac{\alpha}{2} \|\boldsymbol{b}_j\|^2 = (1/2 + 1/q)\alpha^{q/(q+2)} \sum_{j=1}^{r} \sigma_j^{2q/(2+q)}(\boldsymbol{X}). \tag{16}$$

By choosing an appropriate $q$, these representations give arbitrarily tight approximations to the rank, since $\|\boldsymbol{X}\|_{\mathrm{S}_p}^p \to \mathrm{rank}(\boldsymbol{X})$ as $p \to 0$. For example, use (16) in Theorem 2 when $q = \frac{1}{4}$ to see

$$\min_{\sum_{j=1}^d \boldsymbol{a}_j \boldsymbol{b}_j^T = \boldsymbol{X}} \sum_{j=1}^d \frac{1}{1/4} \|\boldsymbol{a}_j\|^{1/4} + \frac{\alpha}{2} \|\boldsymbol{b}_j\|^2 = 4.5 \alpha^{1/9} \sum_{i=1}^d \sigma_i^{2/9}(\boldsymbol{X}) = 4.5 \alpha^{1/9} \|\boldsymbol{X}\|_{\mathrm{S}_{2/9}}^{2/9}. \quad (17)$$

Equality holds in equation (15) when $\boldsymbol{A} = \alpha^{1/(q+1)} \boldsymbol{U}_X \boldsymbol{S}_X^{1/(q+1)}$ and $\boldsymbol{B} = \alpha^{-1/(q+1)} \boldsymbol{S}_X^{q/(q+1)} \boldsymbol{V}_X^T$; in equation (16), when $\boldsymbol{A} = \alpha^{1/(q+2)} \boldsymbol{U}_X \boldsymbol{S}_X^{2/(q+2)}$ and $\boldsymbol{B} = \alpha^{-1/(q+2)} \boldsymbol{S}_X^{q/(q+2)} \boldsymbol{V}_X^T$.

# 4  Application to low-rank matrix completion

As an application, we model noiseless matrix completion using FGSR as a surrogate for the rank:

$$\underset{\boldsymbol{X}}{\mathrm{minimize}} \ \mathrm{FGSR}(\boldsymbol{X}), \quad \text{subject to } P_\Omega(\boldsymbol{X}) = P_\Omega(\boldsymbol{M}). \quad (18)$$

Take $\mathrm{FGSR}_{2/3}$ as an example. We rewrite (18) as

$$\underset{\boldsymbol{X}, \boldsymbol{A}, \boldsymbol{B}}{\mathrm{minimize}} \ \|\boldsymbol{A}\|_{2,1} + \frac{\alpha}{2} \|\boldsymbol{B}\|_F^2, \quad \text{subject to } \boldsymbol{X} = \boldsymbol{A}\boldsymbol{B}, \ P_\Omega(\boldsymbol{X}) = P_\Omega(\boldsymbol{M}). \quad (19)$$

This problem is separable in the three blocks of unknowns $\boldsymbol{X}$, $\boldsymbol{A}$, and $\boldsymbol{B}$. We propose to use the Alternating Direction Method of Multipliers (ADMM) [35, 36, 37] with linearization to solve this problem, as the ADMM subproblem for $\boldsymbol{A}$ has no closed-form solution. Details are in the supplement.

Now consider an application to noisy matrix completion. Suppose we observe $P_\Omega(\boldsymbol{M}_e)$ with $\boldsymbol{M}_e = \boldsymbol{M} + \boldsymbol{E}$, where $\boldsymbol{E}$ represents measurement noise. Model the problem using $\mathrm{FGSR}_{2/3}$ as

$$\underset{\boldsymbol{A}, \boldsymbol{B}}{\mathrm{minimize}} \ \|\boldsymbol{A}\|_{2,1} + \frac{\alpha}{2} \|\boldsymbol{B}\|_F^2 + \frac{\beta}{2} \|P_\Omega(\boldsymbol{M}_e - \boldsymbol{A}\boldsymbol{B})\|_F^2. \quad (20)$$

We can still solve the problem via linearized ADMM. However, proximal alternating linearized minimization (PALM) [38, 39] gives a more efficient method. Details are in the supplement.

Motivated by Theorem 2, we can also model noisy matrix completion with a sharper rank surrogate:

$$\underset{\boldsymbol{A}, \boldsymbol{B}}{\mathrm{minimize}} \ \frac{1}{2} \|\mathcal{P}_\Omega(\boldsymbol{M}_e - \boldsymbol{A}\boldsymbol{B})\|_F^2 + \gamma \Big( \frac{1}{q} \|\boldsymbol{A}\|_{2,q}^q + \frac{\alpha}{2} \|\boldsymbol{B}^T\|_F^2 \Big), \quad (21)$$

where $q \in \{1, \frac{1}{2}, \frac{1}{4}, \cdots\}$ and $\|\boldsymbol{A}\|_{2,q} := \Big( \sum_{j=1}^d \|\boldsymbol{a}_j\|^q \Big)^{1/q}$. When $q < 1$, we suggest solving the problem (21) using PALM coupled with iteratively reweighted minimization [24]. According to the number of degrees of freedom of low-rank matrix, we suggest $d = |\Omega|/(m+n)$ in practical applications.

# 5  Generalization error bound for LRMC

Above, we proposed a method to solve LRMC problems using a FGSR as a rank surrogate. Here, we develop an upper bound on the error of the resulting estimator using a new generalization bound for LRMC with a Schatten-$p$ norm constraint. Similar bounds are available for LRMC using the nuclear norm [30] and max norm [5].

Consider the following observation model. A matrix $\boldsymbol{M}$ is corrupted with iid $\mathcal{N}(0, \epsilon^2)$ noise $\boldsymbol{E}$ to form $\boldsymbol{M}_e = \boldsymbol{M} + \boldsymbol{E}$. Suppose each entry of $\boldsymbol{M}_e$ is observed independently with probability $\rho$ and the number of observed entries is $|\Omega|$, where $\mathbb{E}|\Omega| = \rho mn$.

Choose $q \in \{1, \frac{1}{2}, \frac{1}{4}, \cdots\}$ and $p = \frac{2q}{2+q}$. For any $\gamma > 0$, consider a solution $(\boldsymbol{A}, \boldsymbol{B})$ to (21). Let $\|\boldsymbol{A}\boldsymbol{B}\|_{\mathrm{S}_p}^p = R_p$. Then use Theorem 2 to see that the following problem has the same solution,

$$\underset{\|\boldsymbol{X}\|_{\mathrm{S}_p}^p \le R_p, \mathrm{rank}(\boldsymbol{X}) \le d}{\mathrm{minimize}} \ \|\mathcal{P}_\Omega(\boldsymbol{M}_e - \boldsymbol{X})\|_F^2. \quad (22)$$

Therefore, we may solve (21) using the methods described above to find a solution to (22) efficiently. In this section, we provide generalization error bounds for the solution $\hat{\boldsymbol{M}}$ of (22).

## 5.1 Bound with optimal solution

Without loss of generality, we may assume $\|M\|_\infty \leq \varsigma/\sqrt{mn}$ for some constant $\varsigma$. Hence it is reasonable to assume that $\epsilon = \epsilon_0/\sqrt{mn}$ for some constant $\epsilon_0$. The following theorem provides a generalization error bound for the solution of (22).

**Theorem 3.** *Suppose* $\|M\|_{S_p}^p \leq R_p$, $\hat{M}$ *is the optimal solution of (22), and* $|\Omega| \geq \frac{32}{3} n \log^2 n$. *Denote* $\zeta := \max\{\|M\|_\infty, \|\hat{M}\|_\infty\}$. *Then there exist numerical constants* $c_1$ *and* $c_2$ *such that the following inequality holds with probability at least* $1 - 5n^{-2}$

$$\|M - \hat{M}\|_F^2 \leq \max\left\{ c_1 \zeta^2 \frac{n \log n}{|\Omega|}, (5.5 + \sqrt{10}) R_p \left( (4\sqrt{3}\epsilon_0 + c_2\zeta)^2 \frac{n \log n}{|\Omega|} \right)^{1-p/2} \right\}. \quad (23)$$

When $|\Omega|$ is sufficiently large, we see that the second term in the brace of (23) is the dominant term, which decreases as $p$ decreases. A more complicated but more informative bound can be found in the supplement (inequality (24)). In sum, Theorem 3 shows it is possible to reduce the matrix completion error by using a smaller $p$ in (22) or a smaller $q$ in (21).

## 5.2 Bound with arbitrary *A* and *B*

Since (21) and (22) are nonconvex problems, it is difficult to guarantee that an optimization method has found a globally optimal solution. The following theorem provides a generalization bound for any feasible point $(\hat{A}, \hat{B})$ of (21):

**Theorem 4.** *Suppose* $M_e = M + E$. *For any* $\hat{A}$ *and* $\hat{B}$, *let* $\hat{M} = \hat{A}\hat{B}$ *and* $d$ *be the number of nonzero columns of* $\hat{A}$. *Define* $\zeta := \max\{\|M\|_\infty, \|\hat{M}\|_\infty\}$. *Then there exists a numerical constant* $C_0$, *such that with probability at least* $1 - 2\exp(-n)$, *the following inequality holds:*

$$\frac{\|M - \hat{M}\|_F}{\sqrt{mn}} \leq \frac{\|\mathcal{P}_\Omega(M_e - \hat{M})\|_F}{\sqrt{|\Omega|}} + \frac{\|E\|_F}{\sqrt{mn}} + C_0 \zeta \left( \frac{nd \log n}{|\Omega|} \right)^{1/4}.$$

Theorem 4 indicates that if the training error $\|\mathcal{P}_\Omega(M_e - \hat{A}\hat{B})\|_F$ and the number $d$ of nonzero columns of $\hat{A}$ are small, the matrix completion error is small. In particular, if $E = 0$ and $\mathcal{P}_\Omega(M_e - \hat{A}\hat{B}) = 0$, the matrix completion error is upper-bounded by $C_0 \zeta \left( \frac{nd \log n}{|\Omega|} \right)^{1/4}$. We hope that a smaller $q$ in (21) can lead to smaller training error and $d$ such that the upper bound of matrix completion error is smaller. Indeed, in our experiments, we find that smaller $q$ often leads to smaller matrix completion error, but the improvement saturates quickly as $q$ decreases. We find $q = 1$ or $\frac{1}{2}$ (corresponding to a Schatten-$p$ norm with $p = \frac{2}{3}$ or $\frac{2}{5}$) are enough to provide high matrix completion accuracy and outperform max norm and nuclear norm.

# 6 Application to robust PCA

Suppose a fraction of entries in a matrix are corrupted in random locations. Formally, we observe

$$M_e = M + E, \quad (24)$$

where $M$ is a low-rank matrix and $E$ is the sparse corruption matrix whose nonzero entries may be arbitrary. The robust principal component analysis (RPCA) asks to recover $M$ from $M_e$; a by-now classic approach uses nuclear norm minimization [13]. We propose to use FGSR instead, and solve

$$\underset{A,B,E}{\text{minimize}} \ \frac{1}{q}\|A\|_{2,q}^q + \frac{\alpha}{2}\|B\|_F^2 + \lambda\|E\|_1, \quad \text{subject to } M_e = AB + E, \quad (25)$$

where $q \in \{1, \frac{1}{2}, \frac{1}{4}, \cdots\}$. An optimization algorithm is detailed in the supplement.

# 7 Numerical results

## 7.1 Matrix completion

**Baseline methods** We compare the FGSR regularizers with the nuclear norm, truncated nuclear norm [19], weighted nuclear norm [20], F-nuclear norm, max norm [31], Riemannian pursuit [29],

Schatten-$p$ norm, Bi-nuclear norm [33], and F$^2$+nuclear norm [34]. We choose the parameters of all methods to ensure they perform as well as possible. Details about the optimizations, parameters, evaluation metrics are in the supplement. All experiments present the average of ten trials.

**Noiseless synthetic data**   We generate random matrices of size $500 \times 500$ and rank 50. More details about the experiment are in the supplement. In Figure 1(a), the factored methods all use factors of size $d = 1.5r$. We see the Schatten-$p$ norm ($p = \frac{2}{3}, \frac{1}{2}, \frac{1}{4}$), Bi-nuclear norm, F$^2$+nuclear norm, FGSR$_{2/3}$, and FGSR$_{1/2}$ have similar performances and outperform other methods when the *missing rate* (proportion of unobserved entries) is high. In particular, the F-nuclear norm outperforms the nuclear norm because the bound $d$ on the rank is binding. In Figure 1(b) and (c), in which the missing rates are high, the max norm and F-nuclear norm are sensitive to the initial rank $d$, while the F$^2$+nuclear norm, Bi-nuclear norm, FGSR$_{2/3}$, and FGSR$_{1/2}$ always have nearly zero recovery error. Interestingly, the max norm and F-nuclear norm are robust to the initial rank when the missing rate is much lower than 0.6 in this experiment. In Figure 1(d), we compare the computational time in the case of missing rate$= 0.7$, in which, for fair comparison, the optimization algorithms of all methods were stopped when the relative change of the recovered matrix was less than $10^{-5}$ or the number of iterations reached 1000. The computational cost of nuclear norm, truncated nuclear norm, weighted nuclear norm, and Schatten-$\frac{1}{2}$ norm are especially large, as they require computing the SVD in every iteration. The computational costs of max norm, F-nuclear norm, F$^2$+nuclear norm, and Bi-nuclear norm increase quickly as the initial rank $d$ increases. In contrast, our FGSR$_{2/3}$ and FGSR$_{1/2}$ are very efficient even when the initial rank is large, because they are SVD-free and able to reduce the size of the factors in the progress of optimization. While Riemannian pursuit is a bit faster than FGSR, FGSR has lower error. Note that the Riemannian pursuit code mixes C and MATLAB, while all other methods are written in pure MATLAB, explaining (part of) its more nimble performance.

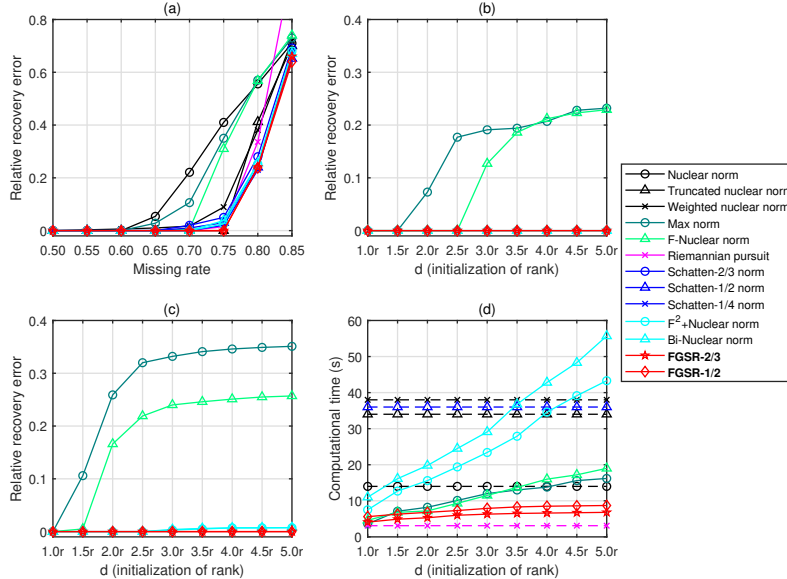

Figure 1: Matrix completion on noiseless synthetic data ($r = 50$): (a) the effect of missing rate on recovery error; (b)(c) the effect of rank initialization on recovery error (missing rate $= 0.6$ or $0.7$); (d) the effect of rank initialization on computational cost (missing rate $= 0.7$).

**Noisy synthetic data**   We simulate a noisy matrix completion problem by adding Gaussian noise to low-rank random matrices. We omit F$^2$+nuclear norm and Bi-nuclear norm from these results because they are less efficient that FGSR$_{2/3}$ and FGSR$_{1/2}$ but perform similarly on recovery error. The recovery errors for different missing rate are reported in Figure 2 (a) and (b) for SNR $= 10$ and SNR $= 5$ (SNR$:= \|\boldsymbol{M}\|_F / \|\boldsymbol{E}\|_F$) respectively. The max norm outperforms the nuclear norm when the missing rate is low. The recovery errors of Schatten-$\frac{1}{2}$ norm, FGSR$_{2/3}$, and FGSR$_{1/2}$ are much lower than those of others. Figure 2(c) demonstrates that our FGSR$_{2/3}$ and FGSR$_{1/2}$ are robust to the initial rank, while max norm and F-nuclear norm degrade as the initial rank increases. In Figure

2(d), we see decreasing $p$ from 1 to 2/9 reduces the recovery error significantly, but the recovery error stabilizes for smaller $p$. This result is consistent with Theorem 3.

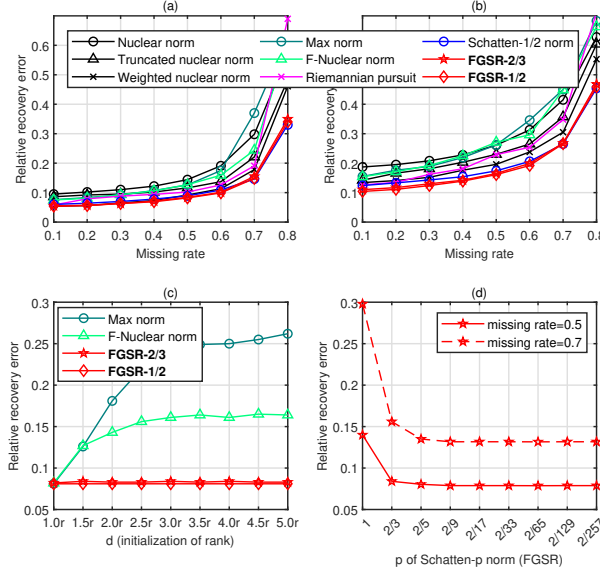

Figure 2: Matrix completion on noisy synthetic data: (a)(b) recovery error when SNR = 10 or 5; (c) the effect of rank initialization on recovery error (SNR = 10, missing rate = 0.5); (d) the effect of $p$ in Schatten-$p$ norm (using FGSR when $p < 1$).

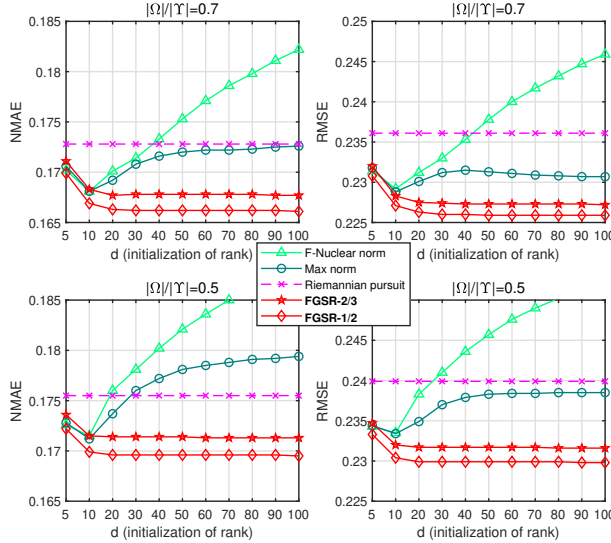

Figure 3: NMAE and RMSE on Movielens-1M data ($\Upsilon$: known entries; $\Omega$: sampled entries from $\Upsilon$)

**Real data**   We consider the MovieLens-1M dataset [40], which consists of 1 million ratings (1 to 5) for 3900 movies by 6040 users. The movies rated by less than 5 users are deleted in this study because the corresponding ratings may never be recovered when the matrix rank is higher than 5. We randomly sample 70% or 50% of the known ratings of each user and perform matrix completion. The normalized mean absolute error (NMAE) [3, 8] and normalized root-mean-squared-error (RMSE) [8] are reported in Figure 3, in which each value is the average of ten repeated trials and the standard deviation is less than 0.0003. Although Riemannian pursuit can adaptively determine the rank, its performance is not satisfactory. As the initial rank increases, the NMAE and RMSE of max norm

and F-nuclear norm increase. In contrast, FGSR$_{2/3}$ and FGSR$_{1/2}$ have consistent low NMAE and RMSE. Moreover, FGSR$_{1/2}$ outperforms FGSR$_{2/3}$.

## 7.2 Robust PCA

We simulate a corrupted matrix as $\boldsymbol{M}_e = \boldsymbol{M} + \boldsymbol{E}$, where $\boldsymbol{M}$ is a random matrix of size $500 \times 500$ with rank 50 and $\boldsymbol{E}$ is a sparse matrix whose nonzero entries are $\mathcal{N}(0, \epsilon^2)$. Define the signal-noise-ratio SNR$_c := \sigma/\epsilon$, where $\sigma$ denotes the standard deviation of the entries of $\boldsymbol{M}$. Figure 4(a) and (b) show the recovery errors for different noise densities (proportion of nonzero entries of $\boldsymbol{E}$). When the noise density is high, FGSR$_{2/3}$ and FGSR$_{1/2}$ outperform nuclear norm and F-nuclear norm. Figure 4(c) and (d) shows again that unlide the F-nuclear norm, FGSR$_{2/3}$ and FGSR$_{1/2}$ are not sensitive to the initial rank, and that FGSR$_{1/2}$ outperforms FGSR$_{2/3}$ slightly when the noise density is high. More results, including for image denoising, appear in the supplement.

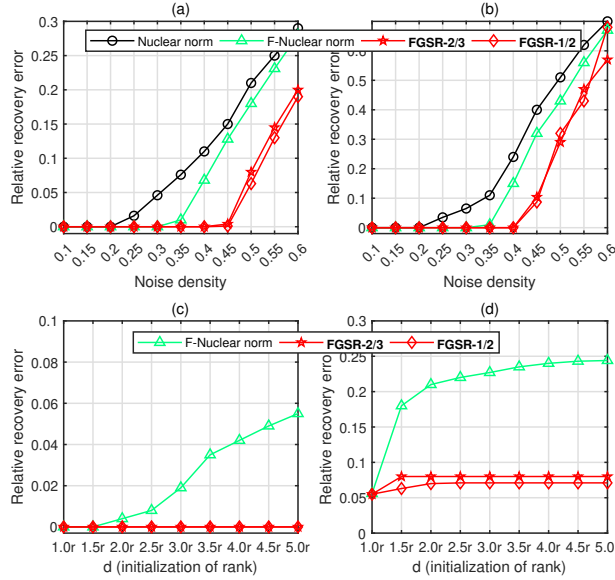

Figure 4: RPCA on synthetic data: (a)(b) recovery error when SNR$_c = 1$ or $0.2$; (c)(d) the effect of rank initialization on recovery error (SNR$_c = 1$, noise density $= 0.3$ or $0.5$).

## 8 Conclusion

This paper proposed a class of nonconvex surrogates for matrix rank that we call Factor Group-Sparse Regularizers (FGSRs). These FGSRs give a factored formulation for certain Schatten-$p$ norms with arbitrarily small $p$. These FGSRs are tighter surrogates for the rank than the nuclear norm, can be optimized without the SVD, and perform well in denoising and completion tasks regardless of the initial choice of rank. In addition, we provide generalization error bounds for LRMC using the FGSR (or, more generally, any Schatten-$p$ norm) as a regularizer. Our experimental results demonstrate the proposed methods[2] achieve state-of-the-art performances in LRMC and RPCA.

These experiments provide compelling evidence that PALM and ADMM may often (perhaps always) converge to the global optimum of these problems. A full convergence theory is an interesting problem for future work. A proof of global convergence would reveal the required sample complexity for LRMC and RPCA with FGSR as a computationally tractable rank proxy.

### Acknowledgements

The authors gratefully acknowledge support from DARPA Award FA8750-17-2-0101 and NSF CCF-1740822.

## Footnotes

[1]Note that formally $\| \cdot \|_{\mathrm{S}p}$ with $0 \le p < 1$ is a quasi-norm, not a norm; abusively, we still use the term "norm" in this paper.

[2]The MATLAB codes of the proposed methods are available at *https://github.com/udellgroup/Codes-of-FGSR-for-effecient-low-rank-matrix-recovery*

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
