[Supplementary Material]

# Factor Group-Sparse Regularization for Efficient Low-Rank Matrix Recovery (supplement)

**Jicong Fan**
Cornell University
Ithaca, NY 14850
jf577@cornell.edu

**Lijun Ding**
Cornell University
Ithaca, NY 14850
ld446@cornell.edu

**Yudong Chen**
Cornell University
Ithaca, NY 14850
yudong.chen@cornell.edu

**Madeleine Udell**
Cornell University
Ithaca, NY 14850
udell@cornell.edu

First of all, let us introduce the following notations. The singular value decompositions of $\boldsymbol{X}$, $\boldsymbol{A}$, and $\boldsymbol{B}$ are

$$\boldsymbol{X} = \boldsymbol{U}_X \boldsymbol{S}_X \boldsymbol{V}_X^T, \ \boldsymbol{A} = \boldsymbol{U}_A \boldsymbol{S}_A \boldsymbol{V}_A^T, \text{and } \boldsymbol{B} = \boldsymbol{U}_B \boldsymbol{S}_B \boldsymbol{V}_B^T$$

respectively. Denote the singular values of $\boldsymbol{X}$, $\boldsymbol{A}$, and $\boldsymbol{B}$ by $\sigma$, $\delta$, and $\theta$ respectively, i.e.

$$\boldsymbol{S}_X = \text{diag}(\sigma_1, \sigma_2, \cdots, \sigma_{r_X}), \ \boldsymbol{S}_A = \text{diag}(\delta_1, \delta_2, \cdots, \delta_{r_A}), \text{and } \boldsymbol{S}_B = \text{diag}(\theta_1, \theta_2, \cdots, \theta_{r_B}).$$

Particularly, $r_X$, $r_A$, and $r_B$ could be larger than the ranks of $\boldsymbol{X}$, $\boldsymbol{A}$, and $\boldsymbol{B}$ respectively, which means the corresponding singular values are zeros. The $\ell_{2,p}$ norm of matrix is defined as $\|\boldsymbol{A}\|_{2,p} := \left( \sum_{j=1}^d \|\boldsymbol{a}_j\|^p \right)^{1/p}$, where $\boldsymbol{A} = [\boldsymbol{a}_1, \boldsymbol{a}_2, \cdots, \boldsymbol{a}_d]$.

## 1 Proof for Theorem 1

**Theorem 1** (main paper, reformulated in the form of matrix norms). *For any matrix $\boldsymbol{X} \in \mathbb{R}^{m \times n}$ with* $\text{rank}(\boldsymbol{X}) = r \leq d \leq \min(m, n)$:
*(a)* $\min_{\boldsymbol{AB}=\boldsymbol{X}} \|\boldsymbol{A}\|_{2,1} + \|\boldsymbol{B}^T\|_{2,1} = 2\|\boldsymbol{X}\|_{\text{S}_{1/2}}^{1/2}$;
*(b)* $\min_{\boldsymbol{AB}=\boldsymbol{X}} \|\boldsymbol{A}\|_{2,1} + \frac{\alpha}{2}\|\boldsymbol{B}\|_F^2 = \frac{3\alpha^{1/3}}{2}\|\boldsymbol{X}\|_{\text{S}_{2/3}}^{2/3}$.

To prove Theorem 1, we need the following lemmas.

**Lemma 1.** *For any matrix $\boldsymbol{A}$, $\|\boldsymbol{A}\|_{2,1} \geq \|\boldsymbol{A}\|_*$.*

*Proof.* Denote the $i$-th column of $\boldsymbol{V}_A^T$ by $\boldsymbol{v}_i$, where $v_{ji}$ is the $j$-th element of $\boldsymbol{v}_i$. Then

$$
\begin{aligned}
\|\boldsymbol{A}\|_{2,1} &= \sum_{i=1}^{d} \sqrt{\boldsymbol{v}_i^T \boldsymbol{S}_A^2 \boldsymbol{v}_i} = \sum_{i=1}^{d} \sqrt{\sum_{j=1}^{r_A} v_{ji}^2 \delta_j^2} \\
&\overset{(i)}{\geq} \sum_{i=1}^{d} \sqrt{\sum_{j=1}^{r_A} (v_{ji}\delta_j)^2 \sum_{j=1}^{r_A} v_{ji}^2} \\
&\overset{(ii)}{\geq} \sum_{i=1}^{d} \sqrt{\left(\sum_{j=1}^{r_A} v_{ji}^2 \delta_j\right)^2} \\
&\overset{(iii)}{=} \sum_{j=1}^{r_A} \delta_j \sum_{i=1}^{d} v_{ji}^2 = \sum_{j=1}^{r_A} \delta_j = \|\boldsymbol{A}\|_*.
\end{aligned}
\tag{1}
$$

Inequality (i) holds because $\sum_{j=1}^{r_A} v_{ji}^2 \leq 1$, $i = 1, 2, \cdots, d$. Inequality (ii) holds due to the Cauchy–Schwarz inequality. In (iii), $\sum_{i=1}^{d} v_{ji}^2 = 1$, $j = 1, 2, \cdots, r_A$. $\qquad\square$

**Lemma 2.** *Suppose $\{h_1, h_2, \cdots, h_l\}$ are arbitrary nonnegative values and denote $\boldsymbol{H} = \mathrm{diag}(h_1, h_2, \cdots, h_l)$. Suppose $0 \leq q \leq 1$ and $\boldsymbol{Q} \in \mathbb{R}^{l \times l}$ is an arbitrary unitary matrix, i.e., $\boldsymbol{Q}\boldsymbol{Q}^T = \boldsymbol{Q}^T\boldsymbol{Q} = \boldsymbol{I}_l$. Then $\mathrm{Tr}(\boldsymbol{H}^{\odot q}) \leq \mathrm{Tr}\big((\boldsymbol{Q}^T\boldsymbol{H}\boldsymbol{Q})^{\odot q}\big)$, where $\odot q$ denotes element-wise power of order $q$.*

*Proof.* Denote the entry $(i, j)$ of $\boldsymbol{Q}$ by $Q_{ij}$, we have $\sum_i Q_{ij}^2 = 1$. Using Jensen's inequality for the concave function, we have $\big(\sum_j Q_{ij}^2 h_j\big)^q \geq \sum_j Q_{ij}^2 h_j^q$. Then

$$
\begin{aligned}
\mathrm{Tr}\big((\boldsymbol{Q}^T\boldsymbol{H}\boldsymbol{Q})^{\odot q}\big) &= \sum_i \Big(\sum_j Q_{ij}^2 h_j\Big)^q \\
&\geq \sum_i \sum_j Q_{ij}^2 h_j^q = \sum_j h_j^q = \mathrm{Tr}(\boldsymbol{H}^{\odot q}).
\end{aligned}
\tag{2}
$$

$\qquad\square$

**Lemma 3.** *Let $\{y_1, y_2, \cdots, y_l\}$ be arbitrary nonnegative values. Suppose $\boldsymbol{P} \in \mathbb{R}^{l \times l}$ and $\boldsymbol{W} \in \mathbb{R}^{l \times l}$ are arbitrary unitary matrices, i.e. $\boldsymbol{P}\boldsymbol{P}^T = \boldsymbol{P}^T\boldsymbol{P} = \boldsymbol{I}_l$ and $\boldsymbol{W}\boldsymbol{W}^T = \boldsymbol{W}^T\boldsymbol{W} = \boldsymbol{I}_l$. Then: (a) $\sum_i \sum_j y_j P_{ji} W_{ij} \leq \sum_j y_j$; (b) $\sum_i \big(\sum_j y_j P_{ji} W_{ij}\big)^2 \leq \sum_j y_j^2$.*

*Proof.* (a) We have $\sum_i P_{ij}^2 = \sum_j P_{ji}^2 = 1$ and $\sum_i W_{ij}^2 = \sum_j W_{ji}^2 = 1$. Then

$$
\begin{aligned}
&\sum_{i=1}^{l} \sum_{j=1}^{l} y_j P_{ji} W_{ij} \\
&\leq \frac{1}{2} \sum_{i=1}^{l} \left(\sum_{j=1}^{l} y_j P_{ji}^2 + \sum_{j=1}^{l} y_j W_{ij}^2\right) \\
&= \frac{1}{2} \sum_{j=1}^{l} y_j \sum_{i=1}^{l} P_{ji}^2 + \frac{1}{2} \sum_{j=1}^{l} y_j \sum_{i=1}^{l} W_{ij}^2 = \sum_{j=1}^{l} y_j.
\end{aligned}
\tag{3}
$$

(b)

$$\sum_{i=1}^{l}\left(\sum_{j=1}^{l} y_j P_{ji} W_{ij}\right)^2$$

$$\overset{(i)}{\leq} \sum_{i=1}^{l}\left(\sum_{j=1}^{l} y_j P_{ji}^2 \sum_{j=1}^{l} y_j W_{ij}^2\right)$$

$$\leq \frac{1}{2}\sum_{i=1}^{l}\left(\left(\sum_{j=1}^{l} y_j P_{ji}^2\right)^2 + \left(\sum_{j=1}^{l} y_j W_{ij}^2\right)^2\right) \tag{4}$$

$$\overset{(ii)}{\leq} \frac{1}{2}\sum_{i=1}^{l}\left(\sum_{j=1}^{l} y_j^2 P_{ji}^2 + \sum_{j=1}^{l} y_j^2 W_{ij}^2\right)$$

$$= \frac{1}{2}\sum_{j=1}^{l} y_j^2 \sum_{i=1}^{l} P_{ji}^2 + \frac{1}{2}\sum_{j=1}^{l} y_j^2 \sum_{i=1}^{l} W_{ij}^2 = \sum_{j=1}^{l} y_j^2.$$

Inequality (i) holds due to Cauchy–Schwarz inequality. Inequality (ii) holds due to Jensen's inequality on the quadratic function, which is convex. □

Enhanced with the lemmas, we can now prove for Theorem 1 as follows.

*Proof.* As $X = AB$, we have $S_X = U_X^T U_A S_A V_A^T U_B S_B V_B^T V_X$ and $r_A = r_B = r_X = d \geq r$. Denote $Q = U_X^T U_A$, $P = V_B^T V_X Q$, and $W = V_A^T U_B$. Because $X$ and $A$ have the same column space while $X$ and $B$ have the same row space, there exist unitary matrices $R_A \in \mathbb{R}^{d \times d}$ and $R_B \in \mathbb{R}^{d \times d}$ such that

$$U_X = U_A R_A \text{ and } V_X = V_B R_B. \tag{5}$$

Then $QQ^T = Q^T Q = I_d$ and $PP^T = P^T P = I_d$. In addition, $WW^T = W^T W = I_d$ because both $V_A$ and $U_B$ are unitary matrices. It follows that $Q^T S_X Q = S_A W S_B P$. In addition, for $1 \leq i \leq d$, $\sum_{j=1}^{d} \theta_j P_{ji} W_{ij} \geq 0$ because the diagonal elements of $W S_B P$ are nonnegative.

(a) We have

$$\|A\|_{2,1} + \|B^T\|_{2,1} \geq 2\sqrt{\|A\|_{2,1}\|B\|_{2,1}}$$

$$\overset{(i)}{\geq} 2\sqrt{\|A\|_*\|B\|_*} = 2\left(\sum_{i=1}^{d}\delta_i \sum_{i=1}^{d}\theta_i\right)^{1/2}$$

$$\overset{(ii)}{\geq} 2\left(\sum_{i=1}^{d}\delta_i \sum_{i=1}^{d}\sum_{j=1}^{d}\theta_j P_{ji} W_{ij}\right)^{1/2}$$

$$\overset{(iii)}{\geq} 2\sum_{i=1}^{d}\delta_i^{1/2}\left(\sum_{j=1}^{d}\theta_j P_{ji} W_{ij}\right)^{1/2} = 2\sum_{i=1}^{d}\left(\sum_{j=1}^{d}\delta_i W_{ij}\theta_j P_{ji}\right)^{1/2} \tag{6}$$

$$= 2\text{Tr}\left((S_A W S_B P)^{\odot 1/2}\right) = 2\text{Tr}\left((Q^T S_X Q)^{\odot 1/2}\right)$$

$$\overset{(iv)}{\geq} 2\text{Tr}\left(S_X^{\odot 1/2}\right) = 2\sum_i \sigma_i^{1/2} = 2\|X\|_{S_{1/2}}^{1/2}.$$

Inequality (i) holds due to Lemma 1. Inequality (ii) holds according to Lemma 3(a). Inequality (iii) holds according to Cauchy–Schwarz inequality. Equality (iv) holds according to Lemma 2. When $A = U_X S_X^{1/2}$ and $B = S_X^{1/2} V_X^T$, the equality holds.

(b) We have

$$\|\boldsymbol{A}\|_{2,1} + \frac{\alpha}{2}\|\boldsymbol{B}^T\|_F^2$$

$$\overset{(i)}{\geq} \frac{1}{2}\|\boldsymbol{A}\|_* + \frac{1}{2}\|\boldsymbol{A}\|_* + \frac{\alpha}{2}\|\boldsymbol{B}^T\|_F^2$$

$$\geq \frac{3\alpha^{1/3}}{2}\left(\|\boldsymbol{A}\|_*\|\boldsymbol{A}\|_*\|\boldsymbol{B}^T\|_F^2\right)^{1/3}$$

$$= \frac{3\alpha^{1/3}}{2}\left(\sum_{i=1}^d \delta_i \sum_{i=1}^d \delta_i \sum_{i=1}^d \theta_i^2\right)^{1/3}$$

$$\overset{(ii)}{\geq} \frac{3\alpha^{1/3}}{2}\left(\sum_{i=1}^d \delta_i \sum_{i=1}^d \delta_i \sum_{i=1}^d \left(\sum_{j=1}^d \theta_j P_{ji} W_{ij}\right)^2\right)^{1/3} \tag{7}$$

$$\overset{(iii)}{\geq} \frac{3\alpha^{1/3}}{2}\sum_{i=1}^d \delta_i^{1/3}\delta_i^{1/3}\left(\sum_{j=1}^d \theta_j P_{ji} W_{ij}\right)^{2/3}$$

$$= \frac{3\alpha^{1/3}}{2}\sum_{i=1}^d \left(\sum_{j=1}^d \delta_i W_{ij}\theta_j P_{ji}\right)^{2/3}$$

$$= \frac{3\alpha^{1/3}}{2}\mathrm{Tr}\left(\left(\boldsymbol{S}_A\boldsymbol{W}\boldsymbol{S}_B\boldsymbol{P}\right)^{\odot 2/3}\right)$$

$$= \frac{3\alpha^{1/3}}{2}\mathrm{Tr}\left(\left(\boldsymbol{Q}^T\boldsymbol{S}_X\boldsymbol{Q}\right)^{\odot 2/3}\right)$$

$$\overset{(iv)}{\geq} \frac{3\alpha^{1/3}}{2}\mathrm{Tr}\left(\boldsymbol{S}_X^{\odot 2/3}\right) = \frac{3\alpha^{1/3}}{2}\sum_i \sigma_i^{2/3} = \frac{3\alpha^{1/3}}{2}\|\boldsymbol{X}\|_{\mathrm{S}_{2/3}}^{2/3}.$$

Inequality (i) holds due to Lemma 1. Inequality (ii) holds according to Lemma 3(b). Inequality (iii) holds according to generalized Hölder's inequality. Equality (iv) holds according to Lemma 2. When $\boldsymbol{A} = \alpha^{1/3}\boldsymbol{U}_X\boldsymbol{S}_X^{2/3}$ and $\boldsymbol{B} = \alpha^{-1/3}\boldsymbol{S}_X^{1/3}\boldsymbol{V}_X^T$, we have $\|\boldsymbol{A}\|_{2,1} = \alpha\|\boldsymbol{B}\|_F^2$ and $\{\boldsymbol{P}, \boldsymbol{W}, \boldsymbol{Q}\}$ are indentity matrices. Then all equalities holds simultaneously. $\qquad\square$

## 2 Proof for Theorem 2

**Theorem 2** (main paper, reformulated in the form of matrix norms). *Fix $\alpha > 0$ and choose $q \in \{1, \frac{1}{2}, \frac{1}{4}, \cdots\}$. For any matrix $\boldsymbol{X} \in \mathbb{R}^{m \times n}$ with $\mathrm{rank}(\boldsymbol{X}) = r \leq d \leq \min(m, n)$,*

*(a)* $\min_{\boldsymbol{AB}=\boldsymbol{X}} \frac{1}{q}\|\boldsymbol{A}\|_{2,q}^q + \alpha\|\boldsymbol{B}^T\|_{2,1} = (1+1/q)\alpha^{q/(q+1)}\|\boldsymbol{X}\|_{\mathrm{S}_{q/(q+1)}}^{q/(q+1)};$

*(b)* $\min_{\boldsymbol{AB}=\boldsymbol{X}} \frac{1}{q}\|\boldsymbol{A}\|_{2,q}^q + \frac{\alpha}{2}\|\boldsymbol{B}^T\|_F^2 = (1/2+1/q)\alpha^{q/(q+2)}\|\boldsymbol{X}\|_{\mathrm{S}_{2q/(2+q)}}^{2q/(2+q)}.$

Before prove Theorem 2, we give the following lemma.

**Lemma 4.** *For any matrix $\boldsymbol{A}$ and $0 < p \leq 1$, $\|\boldsymbol{A}\|_{2,p}^p \geq \|\boldsymbol{A}\|_{S_p}^p$.*

*Proof.* Let $\boldsymbol{A} = \boldsymbol{U}_A\boldsymbol{S}_A\boldsymbol{V}_A^T$ be the SVD of $\boldsymbol{A}$. Denote the $i$-th column of $\boldsymbol{V}_A^T$ by $\boldsymbol{v}_i$, where $v_{ji}$ is the $j$-th element of $\boldsymbol{v}_i$. Then

$$
\begin{aligned}
\|\boldsymbol{A}\|_{2,p}^p &= \sum_{i=1}^{d} \left( \boldsymbol{v}_i^T \boldsymbol{S}_A^2 \boldsymbol{v}_i \right)^{p/2} = \sum_{i=1}^{d} \left( \sum_{j=1}^{r_A} v_{ji}^2 \delta_j^2 \right)^{p/2} \\
&\overset{(i)}{\geq} \sum_{i=1}^{d} \left( \sum_{j=1}^{r_A} (v_{ji}\delta_j)^2 \sum_{j=1}^{r_A} v_{ji}^2 \right)^{p/2} \\
&\overset{(ii)}{\geq} \sum_{i=1}^{d} \left( \sum_{j=1}^{r_A} v_{ji}^2 \delta_j \right)^{p} \\
&\overset{(iii)}{\geq} \sum_{i=1}^{d} \sum_{j=1}^{r_A} v_{ji}^2 \delta_j^p \\
&\overset{(iv)}{=} \sum_{j=1}^{r_A} \delta_j^p \sum_{i=1}^{d} v_{ji}^2 = \sum_{j=1}^{r_A} \delta_j^p = \|\boldsymbol{A}\|_{S_p}^p.
\end{aligned}
\tag{8}
$$

Inequality (i) holds because $\sum_{j=1}^{r_A} v_{ji}^2 \leq 1$, $i = 1, 2, \cdots, d$. Inequality (ii) holds due to the Cauchy–Schwarz inequality. Inequality (iii) holds due to the Jensen's inequality for concave function $x^p$ ($0 < p \leq 1$). In (iv), $\sum_{i=1}^{d} v_{ji}^2 = 1$, $j = 1, 2, \cdots, r_A$. $\qquad \square$

Then let's prove Theorem 2.

*Proof.* (a) We have

$$
\begin{aligned}
&\frac{1}{q}\|\boldsymbol{A}\|_{2,q}^q + \|\boldsymbol{B}^T\|_{2,1} \\
\overset{(i)}{\geq}\ & \underbrace{\|\boldsymbol{A}\|_{S_q}^q + \cdots + \|\boldsymbol{A}\|_{S_q}^q}_{1/q \text{ terms}} + \alpha\|\boldsymbol{B}^T\|_{2,1} \\
\geq\ & (1+1/q)\alpha^{q/(q+1)}\big(\underbrace{\|\boldsymbol{A}\|_{S_q}^q \cdots \|\boldsymbol{A}\|_{S_q}^q}_{1/q \text{ terms}}\|\boldsymbol{B}^T\|_{2,1}\big)^{q/(q+1)} \\
=\ & (1+1/q)\alpha^{q/(q+1)}\bigg(\underbrace{\sum_{i=1}^{d}\delta_i^q \cdots \sum_{i=1}^{d}\delta_i^q}_{1/q \text{ terms}}\sum_{i=1}^{d}\theta_i\bigg)^{q/(q+1)} \\
\overset{(ii)}{\geq}\ & (1+1/q)\alpha^{q/(q+1)}\bigg(\underbrace{\sum_{i=1}^{d}\delta_i^q \cdots \sum_{i=1}^{d}\delta_i^q}_{1/q \text{ terms}}\sum_{i=1}^{d}\Big(\sum_{j=1}^{d}\theta_j P_{ji}W_{ij}\Big)\bigg)^{q/(q+1)} \\
\overset{(iii)}{\geq}\ & (1+1/q)\alpha^{q/(q+1)}\sum_{i=1}^{d}\underbrace{\delta_i^{q/(q^{-1}+1)}\cdots\delta_i^{q/(q^{-1}+1)}}_{1/q \text{ terms}}\Big(\sum_{j=1}^{d}\theta_j P_{ji}W_{ij}\Big)^{1/(q^{-1}+1)} \\
=\ & (1+1/q)\alpha^{q/(q+1)}\sum_{i=1}^{d}\Big(\sum_{j=1}^{d}\delta_i W_{ij}\theta_j P_{ji}\Big)^{1/(q^{-1}+1)} \\
=\ & (1+1/q)\alpha^{q/(q+1)}\text{Tr}\Big(\big(\boldsymbol{S}_A\boldsymbol{W}\boldsymbol{S}_B\boldsymbol{P}\big)^{\odot 1/(q^{-1}+1)}\Big) \\
=\ & (1+1/q)\alpha^{q/(q+1)}\text{Tr}\Big(\big(\boldsymbol{Q}^T\boldsymbol{S}_X\boldsymbol{Q}\big)^{\odot q/(q+1)}\Big) \\
\overset{(iv)}{\geq}\ & (1+1/q)\alpha^{q/(q+1)}\text{Tr}\Big(\boldsymbol{S}_X^{\odot q/(q+1)}\Big) \\
=\ & (1+1/q)\alpha^{q/(q+1)}\sum_i \sigma_i^{q/(q+1)} = (1+1/q)\alpha^{q/(q+1)}\|\boldsymbol{X}\|_{S_{q/(q+1)}}^{q/(q+1)}.
\end{aligned}
\tag{9}
$$

Inequality (i) holds due to Lemma 4. Inequality (ii) holds according to Lemma 3(a). Inequality (iii) holds according to generalized Hölder's inequality. Equality (iv) holds according to Lemma 2. When $\boldsymbol{A} = \alpha^{1/(q+1)}\boldsymbol{U}_X\boldsymbol{S}_X^{1/(q+1)}$ and $\boldsymbol{B} = \alpha^{-1/(q+1)}\boldsymbol{S}_X^{q/(q+1)}\boldsymbol{V}_X^T$, we have $\|\boldsymbol{A}\|_{S_q}^q = \alpha\|\boldsymbol{B}^T\|_{2,1}$ and $\{\boldsymbol{P},\boldsymbol{W},\boldsymbol{Q}\}$ are indentity matrices. Then all equalities holds simultaneously.

(b) We have

$$\frac{1}{q}\|\boldsymbol{A}\|_{2,q}^q + \frac{\alpha}{2}\|\boldsymbol{B}^T\|_F^2$$

$$\stackrel{(i)}{\geq} \underbrace{\frac{1}{2}\|\boldsymbol{A}\|_{S_q}^q + \cdots + \frac{1}{2}\|\boldsymbol{A}\|_{S_q}^q}_{2/q \text{ terms}} + \frac{\alpha}{2}\|\boldsymbol{B}^T\|_F^2$$

$$\geq (1/2 + 1/q)\alpha^{q/(q+2)}\big(\underbrace{\|\boldsymbol{A}\|_{S_q}^q \cdots \|\boldsymbol{A}\|_{S_q}^q}_{2/q \text{ terms}}\|\boldsymbol{B}^T\|_F^2\big)^{q/(q+2)}$$

$$= (1/2 + 1/q)\alpha^{q/(q+2)}\bigg(\underbrace{\sum_{i=1}^d \delta_i^q \cdots \sum_{i=1}^d \delta_i^q}_{2/q \text{ terms}} \sum_{i=1}^d \theta_i^2\bigg)^{q/(q+2)}$$

$$\stackrel{(ii)}{\geq} (1/2 + 1/q)\alpha^{q/(q+2)}\bigg(\underbrace{\sum_{i=1}^d \delta_i^q \cdots \sum_{i=1}^d \delta_i^q}_{2/q \text{ terms}} \sum_{i=1}^d \Big(\sum_{j=1}^d \theta_j P_{ji} W_{ij}\Big)^2\bigg)^{q/(q+2)} \tag{10}$$

$$\stackrel{(iii)}{\geq} (1/2 + 1/q)\alpha^{q/(q+2)} \sum_{i=1}^d \underbrace{\delta_i^{q/(2q^{-1}+1)} \cdots \delta_i^{q/(2q^{-1}+1)}}_{2/q \text{ terms}} \Big(\sum_{j=1}^d \theta_j P_{ji} W_{ij}\Big)^{2/(2q^{-1}+1)}$$

$$= (1/2 + 1/q)\alpha^{q/(q+2)} \sum_{i=1}^d \Big(\sum_{j=1}^d \delta_i W_{ij} \theta_j P_{ji}\Big)^{2q/(q+2)}$$

$$= (1/2 + 1/q)\alpha^{q/(q+2)} \text{Tr}\Big(\big(\boldsymbol{S}_A \boldsymbol{W} \boldsymbol{S}_B \boldsymbol{P}\big)^{\odot 2q/(q+2)}\Big)$$

$$= (1/2 + 1/q)\alpha^{q/(q+2)} \text{Tr}\Big(\big(\boldsymbol{Q}^T \boldsymbol{S}_X \boldsymbol{Q}\big)^{\odot 2q/(q+2)}\Big)$$

$$\stackrel{(iv)}{\geq} (1/2 + 1/q)\alpha^{q/(q+2)} \text{Tr}\Big(\boldsymbol{S}_X^{\odot 2q/(q+2)}\Big)$$

$$= (1/2 + 1/q)\alpha^{q/(q+2)} \sum_i \sigma_i^{2q/(q+2)} = (1/2 + 1/q)\alpha^{q/(q+2)}\|\boldsymbol{X}\|_{S_{2q/(2+q)}}^{2q/(2+q)}.$$

Inequality (i) holds due to Lemma 4. Inequality (ii) holds according to Lemma 3(b). Inequality (iii) holds according to generalized Hölder's inequality. Equality (iv) holds according to Lemma 2. When $\boldsymbol{A} = \alpha^{1/(q+2)}\boldsymbol{U}_X \boldsymbol{S}_X^{2/(2+q)}$ and $\boldsymbol{B} = \alpha^{-1/(q+2)}\boldsymbol{S}_X^{q/(2+q)}\boldsymbol{V}_X^T$, we have $\|\boldsymbol{A}\|_{S_q}^q = \alpha\|\boldsymbol{B}\|_F^2$ and $\{\boldsymbol{P}, \boldsymbol{W}, \boldsymbol{Q}\}$ are indentity matrices. Then all equalities holds simultaneously. $\square$

## 3  Proof for Theorem 3

**Theorem 3.** *Suppose* $\|M\|_{S_p}^p \leq R_p$, $\hat{M}$ *is the optimal solution of (20), and* $|\Omega| \geq \frac{32}{3}n\log^2 n$. *Denote* $\zeta := \max\{\|M\|_\infty, \|\hat{M}\|_\infty\}$. *Then there exist numerical constants* $c_1$ *and* $c_2$ *such that the following inequality holds with probability at least* $1 - 5n^{-2}$

$$\|\boldsymbol{M} - \hat{\boldsymbol{M}}\|_F^2 \leq \max\left\{ c_1\zeta^2\frac{n\log n}{|\Omega|}, (5.5 + \sqrt{10})R_p\left((4\sqrt{3}\epsilon_0 + c_2\zeta)^2\frac{n\log n}{|\Omega|}\right)^{1-p/2}\right\}. \tag{11}$$

Before prove the above theorem, we introduce a restricted strong convexity (RSC) result:

**Lemma 5.** *Let* $\mathcal{P}_\Omega$ *be the random sampling operator. Then there are universal positive constants* $(c_1', c_2')$ *such that*

$$\left|\frac{mn}{|\Omega|}\|\mathcal{P}_\Omega(\boldsymbol{\Delta})\|_F^2 - \|\boldsymbol{\Delta}\|_F^2\right| \leq c_1'\sqrt{mn}\|\boldsymbol{\Delta}\|_\infty\|\boldsymbol{\Delta}\|_*\sqrt{\frac{n\log n}{|\Omega|}} + c_2'mn\|\boldsymbol{\Delta}\|_\infty^2\frac{n\log n}{|\Omega|} \tag{12}$$

*for all* $\boldsymbol{\Delta} \in \mathbb{R}^{m \times n} (m \leq n)$, *uniformly with probability at least* $1 - 2e^{-\frac{1}{2}n\log n}$.

Our RSC is similar to the one in [1]. One major difference compared to [1] is that here $\Omega$ is sampled without replacement from $[m] \times [n]$.

We define the set

$$\mathbb{S}_p(R_p) = \left\{ \boldsymbol{Y} \in \mathbb{R}^{m \times n} \mid \sum_{i=1}^{\min\{m,n\}} |\sigma_i(\boldsymbol{Y})|^p \leq R_p \right\}, \tag{13}$$

where $\sigma_i(\boldsymbol{Y})$ denotes the $i$-th singular value of $\boldsymbol{Y}$ and $0 \leq p \leq 1$. Then we have the following lemma:

**Lemma 6.** *For any matrix* $\boldsymbol{\Delta} \in \mathbb{S}_p(2R_p)$ *and any positive number* $\tau$, *the following inequality holds*

$$\|\boldsymbol{\Delta}\|_* \leq \sqrt{2R_p}\tau^{-p/2}\|\boldsymbol{\Delta}\|_F + 2R_p\tau^{1-p}. \tag{14}$$

**Lemma 7.** *Suppose the entries of* $\boldsymbol{E} \in \mathbb{R}^{m \times n}$ $(m \leq n)$ *are drawn from* $\mathcal{N}(0, \epsilon^2)$ *and* $|\Omega| \geq \frac{32}{3}n\log^2 n$. *Then*

$$\mathbb{P}\left(\|\mathcal{P}_\Omega(\boldsymbol{E})\|_2 \leq 2\sqrt{3}\epsilon\sqrt{\frac{|\Omega|\log n}{m}}\right) \geq 1 - 4n^{-2}. \tag{15}$$

Now we prove Theorem 3.

*Proof.* The optimal $\hat{\boldsymbol{M}}$ indicates

$$\|\mathcal{P}_\Omega(\boldsymbol{M}_e - \hat{\boldsymbol{M}})\|_F^2 \leq \|\mathcal{P}_\Omega(\boldsymbol{M}_e - \boldsymbol{M})\|_F^2, \tag{16}$$

and

$$\|\mathcal{P}_\Omega(\boldsymbol{M} - \hat{\boldsymbol{M}})\|_F^2 + 2\langle\mathcal{P}_\Omega(\boldsymbol{M} - \hat{\boldsymbol{M}}), \mathcal{P}_\Omega(\boldsymbol{E})\rangle \leq 0. \tag{17}$$

Denote $\boldsymbol{\Delta} := \boldsymbol{M} - \hat{\boldsymbol{M}}$. Then $\boldsymbol{\Delta} \in \mathbb{S}_p(2R_p)$. We have

$$\|\mathcal{P}_\Omega(\boldsymbol{\Delta})\|_F^2 \leq 2|\langle\mathcal{P}_\Omega(\boldsymbol{\Delta}), \mathcal{P}_\Omega(\boldsymbol{E})\rangle| \leq 2\|\boldsymbol{\Delta}\|_*\|\mathcal{P}_\Omega(\boldsymbol{E})\|_2 \tag{18}$$

In addition, we have

$$\|\boldsymbol{\Delta}\|_\infty \leq 2\zeta/\sqrt{mn}. \tag{19}$$

Then according to Lemma 5, the following inequality holds with probability at least $1 - 2e^{-\frac{1}{2}n\log n}$

$$\|\boldsymbol{\Delta}\|_F^2 \leq \frac{mn}{|\Omega|}\|\mathcal{P}_\Omega(\boldsymbol{\Delta})\|_F^2 + 2c_1'\zeta\|\boldsymbol{\Delta}\|_*\sqrt{\frac{n\log n}{|\Omega|}} + 4c_2'\zeta^2\frac{n\log n}{|\Omega|}. \tag{20}$$

Using (18), (20), Lemma 7, and Lemma 6, we obtain

$$\begin{aligned}
\|\boldsymbol{\Delta}\|_F^2 &\leq \left(\frac{2mn}{|\Omega|}\|\mathcal{P}_\Omega(\boldsymbol{E})\|_2 + 2c_1'\zeta\sqrt{\frac{n\log n}{|\Omega|}}\right)\|\boldsymbol{\Delta}\|_* + 4c_2'\zeta^2\frac{n\log n}{|\Omega|} \\
&\leq \left(4\sqrt{3}n\epsilon\sqrt{\frac{m\log n}{|\Omega|}} + 2c_1'\zeta\sqrt{\frac{n\log n}{|\Omega|}}\right)\|\boldsymbol{\Delta}\|_* + 4c_2'\zeta^2\frac{n\log n}{|\Omega|} \\
&\leq \sqrt{2R_p}z\tau^{-p/2}\|\boldsymbol{\Delta}\|_F + 2R_p z\tau^{1-p} + 4c_2'\zeta^2\frac{n\log n}{|\Omega|},
\end{aligned} \tag{21}$$

where $z = 4\sqrt{3}n\epsilon\sqrt{\frac{m\log n}{|\Omega|}} + 2c_1'\zeta\sqrt{\frac{n\log n}{|\Omega|}}$. Let $\tau = z$, we have

$$\|\boldsymbol{\Delta}\|_F^2 - \sqrt{2R_p}\tau^{1-p/2}\|\boldsymbol{\Delta}\|_F - 2R_p\tau^{2-p} - 4c_2'\zeta^2\frac{n\log n}{|\Omega|} \leq 0. \tag{22}$$

Solving the inequality yields

$$\|\boldsymbol{\Delta}\|_F \leq \frac{1}{2}\left(\sqrt{2R_p}\tau^{1-p/2} + \sqrt{10R_p\tau^{2-p} + 16c_2'\zeta^2\frac{n\log n}{|\Omega|}}\right). \tag{23}$$

Note that $\epsilon = \epsilon_0/\sqrt{mn}$, we have $\tau = (4\sqrt{3}\epsilon_0 + 2c_1'\zeta)\sqrt{\dfrac{n\log n}{|\Omega|}}$. It follows from (23) that

$$
\|\mathbf{\Delta}\|_F^2 \le \frac{1}{4}\Bigg( \sqrt{2R_p}\Big((4\sqrt{3}\epsilon_0 + 2c_1'\zeta)\sqrt{\frac{n\log n}{|\Omega|}}\Big)^{1-p/2}
$$
$$
+ \sqrt{10R_p\Big((4\sqrt{3}\epsilon_0 + 2c_1'\zeta)\sqrt{\frac{n\log n}{|\Omega|}}\Big)^{2-p} + 16c_2'\zeta^2\frac{n\log n}{|\Omega|}} \Bigg)^2. \tag{24}
$$

It indicates that for sufficiently large $|\Omega|$, smaller $p$ leads to smaller upper bound of $\|\mathbf{M} - \hat{\mathbf{M}}\|_F^2$ but the decreasing may not be significant when $p$ approaches to zero.

To make a simpler formulation, we consider the following cases. If $2R_p\tau^{2-p} \le \dfrac{16}{5}c_2'\zeta^2\dfrac{n\log n}{|\Omega|}$, we have

$$
\|\mathbf{\Delta}\|_F^2 \le (8.8 + 8\sqrt{2/5})c_2'\zeta^2\frac{n\log n}{|\Omega|}. \tag{25}
$$

Otherwise, we have

$$
\|\mathbf{\Delta}\|_F^2 \le (5.5 + \sqrt{10})R_p\tau^{2-p}. \tag{26}
$$

Putting above results together, with probability at least $1 - 5n^{-2}$, we obtain

$$
\|\mathbf{\Delta}\|_F^2 \le \max\left\{ c_1\zeta^2\frac{n\log n}{|\Omega|}, (5.5 + \sqrt{10})R_p\Big(4\sqrt{3}n\epsilon\sqrt{\frac{m\log n}{|\Omega|}} + c_2\zeta\sqrt{\frac{n\log n}{|\Omega|}}\Big)^{2-p} \right\}, \tag{27}
$$

where $c_1 = (8.8 + 8\sqrt{2/5})c_2'$ and $c_2 = 2c_1'$. We have

$$
\|\mathbf{M} - \hat{\mathbf{M}}\|_F^2 \le \max\left\{ c_1\zeta^2\frac{n\log n}{|\Omega|}, (5.5 + \sqrt{10})R_p\Big((4\sqrt{3}\epsilon_0 + c_2\zeta)^2\frac{n\log n}{|\Omega|}\Big)^{1-p/2} \right\}. \tag{28}
$$

$\square$

## 4 Proof for Theorem 4

**Theorem 4.** *Suppose $\mathbf{M}_e = \mathbf{M} + \mathbf{E}$. For any $\hat{\mathbf{A}}$ and $\hat{\mathbf{B}}$, let $\hat{\mathbf{M}} = \hat{\mathbf{A}}\hat{\mathbf{B}}$ and $d$ be the number of nonzero columns of $\hat{\mathbf{A}}$. Define $\zeta := \max\{\|\mathbf{M}\|_\infty, \|\hat{\mathbf{M}}\|_\infty\}$. Then there exists a numerical constant $C_0$, such that with probability at least $1 - 2\exp(-n)$, the following inequality holds:*

$$
\frac{\|\mathbf{M} - \hat{\mathbf{M}}\|_F}{\sqrt{mn}} \le \frac{\|\mathcal{P}_\Omega(\mathbf{M}_e - \hat{\mathbf{M}})\|_F}{\sqrt{|\Omega|}} + \frac{\|\mathbf{E}\|_F}{\sqrt{mn}} + C_0\zeta\Big(\frac{nd\log n}{|\Omega|}\Big)^{1/4}.
$$

First, we reformulate the Theorem 2 of [2] as

**Lemma 8.** *Let $\mathcal{L}_\Omega(\hat{\mathbf{X}}) = \frac{1}{\sqrt{|\Omega|}}\|\mathcal{P}_\Omega(\mathbf{M}_e - \hat{\mathbf{X}})\|_F$ and $\mathcal{L}(\hat{\mathbf{X}}) = \frac{1}{\sqrt{mn}}\|\mathbf{M}_e - \hat{\mathbf{X}}\|_F$ be the empirical and actual loss function respectively. Furthermore, assume entry-wise constraint $\max_{ij}|\hat{\mathbf{X}}_{ij}| \le C_X$. Then for all rank-$r$ matrices $\hat{\mathbf{X}}$, with probability greater than $1 - 2\exp(-n)$, there exists a fixed constant $C_0$ such that*

$$
\sup_{\hat{\mathbf{X}} \in S_r} \Big|\mathcal{L}_\Omega(\hat{\mathbf{X}}) - \mathcal{L}(\hat{\mathbf{X}})\Big| \le C_0 C_X\Big(\frac{nr\log n}{|\Omega|}\Big)^{1/4}.
$$

We have

$$
\frac{\|\mathbf{M}_e - \hat{\mathbf{A}}\hat{\mathbf{B}}\|_F}{\sqrt{mn}}
$$
$$
\le \left| \frac{\|\mathbf{M}_e - \hat{\mathbf{A}}\hat{\mathbf{B}}\|_F}{\sqrt{mn}} - \frac{\|\mathcal{P}_\Omega(\mathbf{M}_e - \hat{\mathbf{A}}\hat{\mathbf{B}})\|_F}{\sqrt{|\Omega|}} \right| + \frac{\|\mathcal{P}_\Omega(\mathbf{M}_e - \hat{\mathbf{A}}\hat{\mathbf{B}})\|_F}{\sqrt{|\Omega|}} \tag{29}
$$
$$
:= R(\hat{\mathbf{A}}, \hat{\mathbf{B}}, \Omega) + \frac{\|\mathcal{P}_\Omega(\mathbf{M}_e - \hat{\mathbf{A}}\hat{\mathbf{B}})\|_F}{\sqrt{|\Omega|}}.
$$

According to Lemma 8, then with probability at least $1 - 2\exp(-n)$, there exists a fixed constant $C_0$ such that

$$\sup_{\hat{\boldsymbol{A}}\hat{\boldsymbol{B}} \in S_d} R(\hat{\boldsymbol{A}}, \hat{\boldsymbol{B}}, \Omega) \leq C_0 \zeta \Big(\frac{nd\log n}{|\Omega|}\Big)^{1/4}. \tag{30}$$

Using (29) and (30), we have

$$
\begin{aligned}
&\frac{\|\boldsymbol{M} - \hat{\boldsymbol{A}}\hat{\boldsymbol{B}}\|_F}{\sqrt{mn}} \\
\leq & \frac{\|\boldsymbol{E}\|_F}{\sqrt{mn}} + \frac{\|\boldsymbol{M}_e - \hat{\boldsymbol{A}}\hat{\boldsymbol{B}}\|_F}{\sqrt{mn}} \\
\leq & \frac{\|\boldsymbol{E}\|_F}{\sqrt{mn}} + C_0\zeta\Big(\frac{nd\log n}{|\Omega|}\Big)^{1/4} + \frac{\|\mathcal{P}_\Omega(\boldsymbol{M}_e - \hat{\boldsymbol{A}}\hat{\boldsymbol{B}})\|_F}{\sqrt{|\Omega|}}.
\end{aligned}
\tag{31}
$$

This finished the proof.

## 5 Proof for Lemmas 5, 6, and 7

### 5.1 Proof for Lemma 5

Before prove Lemma 5, we introduce

**Lemma 9** (Theorem 1.1 in [3]). *Let* $X_1, X_2, \cdots, X_s$ *be independent random variables with values in a measurable space* $(S, \mathcal{B})$ *and let* $\mathcal{F}$ *be a countable class of measurable functions* $f : S \to [-a, a]$, *such that for all* $i$, $\mathbb{E}[f(X_i)] = 0$. *Consider the random variable* $Z = \sup_{f \in \mathcal{F}} \sum_{i=1}^{s} f(X_i)$. *Then for all* $t \geq 0$,

$$\mathbb{P}(Z \geq \mathbb{E}[Z] + t) \leq \exp(-\frac{t^2}{2(\sigma^2 + 2a\mathbb{E}[Z]) + 3at})),$$

*where* $\sigma^2 = \sup_{f \in \mathcal{F}} \sum_{i=1}^{s} \mathbb{E}[f^2(X_i)]$.

The above lemma indicates that there are universal positive constants $c_1$ and $c_2$ such that for any $\epsilon > 0$

$$\mathbb{P}(Z \geq (1+\epsilon)\mathbb{E}[Z] + c_1\sigma\sqrt{t'} + (c_2 + c_1^2/\epsilon)at') \leq e^{-t'}. \tag{32}$$

Now we prove Theorem 3. Without loss of generality, we assume $\|\boldsymbol{\Delta}\|_F^2 = 1$. Then for given constants $\zeta$ and $\theta$, we define the set

$$\mathbb{S}(\zeta, \theta) := \Big\{\boldsymbol{\Delta} \in \mathbb{R}^{m \times n} \mid \|\boldsymbol{\Delta}\|_F^2 = 1, \|\boldsymbol{\Delta}\|_\infty \leq \frac{\zeta}{\sqrt{mn}}, \|\boldsymbol{\Delta}\|_* \leq \theta\Big\} \tag{33}$$

and random variable

$$Z(\zeta, \theta) := \sup_{\boldsymbol{\Delta} \in \mathbb{S}(\zeta, \theta)} \Big| \frac{mn}{|\Omega|} \|\mathcal{P}_\Omega(\boldsymbol{\Delta})\|_F^2 - 1 \Big|. \tag{34}$$

According to $\mathcal{P}_\Omega$, we define $f_{\boldsymbol{\Delta}}(\boldsymbol{S}_k) = \frac{mn}{|\Omega|}\langle\boldsymbol{\Delta}, \boldsymbol{S}_k\rangle^2$, where $\boldsymbol{S}_k = \boldsymbol{e}_{i_k}\boldsymbol{e}_{j_k}^T$ and $k = 1, 2, \cdots, |\Omega|$. $\{f_{\boldsymbol{\Delta}}(\boldsymbol{S}_k)\}$ are independently but not identically distributed. Then we write

$$Z(\zeta, \theta) = \sup_{\boldsymbol{\Delta} \in \mathbb{S}(\zeta, \theta)} \Big| \sum_{k=1}^{|\Omega|} f_{\boldsymbol{\Delta}}(\boldsymbol{S}_k) - \mathbb{E}[f_{\boldsymbol{\Delta}}(\boldsymbol{S}_k)] \Big|. \tag{35}$$

Denote $\bar{f}_{\boldsymbol{\Delta}}(\boldsymbol{S}_k) := f_{\boldsymbol{\Delta}}(\boldsymbol{S}_k) - \mathbb{E}[f_{\boldsymbol{\Delta}}(\boldsymbol{S}_k)]$. Since $0 \leq f_{\boldsymbol{\Delta}}(\boldsymbol{S}) \leq \zeta^2/|\Omega|$, we have $\mathbb{E}[\bar{f}_{\boldsymbol{\Delta}}^2(\boldsymbol{S}_k)] \leq \mathbb{E}[f_{\boldsymbol{\Delta}}^2(\boldsymbol{S}_k)] \leq \zeta^2/|\Omega|\mathbb{E}[f_{\boldsymbol{\Delta}}(\boldsymbol{S}_k)] = \zeta^2/|\Omega|^2$. Then put $Z(\zeta, \theta)$ into (32) and let $\epsilon = 1$ and $t' = n\log n$, where $\sigma^2 \leftarrow \zeta^2/|\Omega|$, $s \leftarrow |\Omega|$, and $a \leftarrow \zeta^2/|\Omega|$. We conclude that there are universal constant $c_1'$ and $c_2'$ such that

$$\mathbb{P}\Big(Z(\zeta, \theta) \geq 2\mathbb{E}[Z(\zeta, \theta)] + \frac{c_1'}{8}\zeta\sqrt{\frac{n\log n}{|\Omega|}} + \frac{c_2'}{4}\zeta^2\frac{n\log n}{|\Omega|}\Big) \leq e^{-n\log n}. \tag{36}$$

Let $\epsilon$ be Rademacher variable, we have

$$\mathbb{E}[Z(\zeta,\theta)] \leq 2\mathbb{E}\left[\sup_{\boldsymbol{\Delta}\in\mathbb{S}(\zeta,\theta)}\left|\sum_{k=1}^{|\Omega|}\epsilon_k\frac{mn}{|\Omega|}\langle\boldsymbol{\Delta},\boldsymbol{S}_k\rangle^2\right|\right] \leq 4\zeta\mathbb{E}\left[\sup_{\boldsymbol{\Delta}\in\mathbb{S}(\zeta,\theta)}\left|\frac{\sqrt{mn}}{|\Omega|}\sum_{k=1}^{|\Omega|}\epsilon_k\langle\boldsymbol{\Delta},\boldsymbol{S}_k\rangle\right|\right],$$
(37)

where the first and second inequalities follow from the Proposition 4.11 and inequality 5.61 of [4]. Using Hölder's inequality and the fact $\|\boldsymbol{\Delta}\|_* \leq \theta$, we have

$$\mathbb{E}\left[\sup_{\boldsymbol{\Delta}\in\mathbb{S}(\zeta,\theta)}\left|\frac{\sqrt{mn}}{|\Omega|}\sum_{k=1}^{|\Omega|}\epsilon_k\langle\boldsymbol{\Delta},\boldsymbol{S}_k\rangle\right|\right] \leq \theta\mathbb{E}\left[\|\frac{\sqrt{mn}}{|\Omega|}\sum_{k=1}^{|\Omega|}\epsilon_k\boldsymbol{S}_k\|_2\right].$$
(38)

Note that $\mathbb{E}[\epsilon_k\boldsymbol{S}_k] = \boldsymbol{0}$, $\|\epsilon_k\boldsymbol{S}_k\|_2 = 1$, and $\max\{\|\sum_{k=1}^{|\Omega|}\mathbb{E}[\epsilon_k^2\boldsymbol{S}_k\boldsymbol{S}_k^T]\|_2, \|\sum_{k=1}^{|\Omega|}\mathbb{E}[\epsilon_k^2\boldsymbol{S}_k^T\boldsymbol{S}_k]\|_2\} = \frac{|\Omega|}{m}$, where we have assumed $m \leq n$. Then use matrix Bernstein inequality, we obtain

$$\mathbb{P}\left(\|\sum_{k=1}^{|\Omega|}\epsilon_k\boldsymbol{S}_k\|_2 \geq t\right) \leq 2n\exp\left(\frac{-t^2}{2(\frac{|\Omega|}{m}+t/3)}\right).$$

It follows that

$$\mathbb{E}\left[\|\sum_{k=1}^{|\Omega|}\epsilon_k\boldsymbol{S}_k\|_2\right] \leq 2\sqrt{\frac{|\Omega|}{m}}(\sqrt{\pi}+\sqrt{\log 2n}) + 4/3(1+\log 2n) \leq c_t\sqrt{\frac{|\Omega|\log n}{m}},$$
(39)

where we have used the fact $|\Omega| > n\log n$ and $c_t$ is some constant. Combining (37), (38), and (39), we obtain

$$\mathbb{E}[Z(\zeta,\theta)] \leq 4\zeta\theta\frac{\sqrt{mn}}{|\Omega|}c_t\sqrt{\frac{|\Omega|\log n}{m}} \leq \frac{c_1'}{16}\zeta\theta\sqrt{\frac{n\log n}{|\Omega|}},$$
(40)

where $c_1'$ is an appropriate constant. Invoking (40) into (36) and using $\theta \geq 1$, we have

$$\mathbb{P}\left(Z(\zeta,\theta) \geq \frac{c_1'}{4}\zeta\theta\sqrt{\frac{n\log n}{|\Omega|}} + \frac{c_2'}{4}\zeta^2\frac{n\log n}{|\Omega|}\right) \leq e^{-n\log n}.$$
(41)

Note that (41) involves the fixed $\zeta$ and $\theta$. We need to extend it to arbitrary $\sqrt{mn}\|\boldsymbol{\Delta}\|_\infty$ and $\|\boldsymbol{\Delta}\|_*$. The remaining proof is nearly the same as that in Chapter 10 of [4]. We include it here for completeness.

Let $\mathbb{B}_F(1)$ denote the Frobenius ball of norm one in $\mathbb{R}^{m\times n}$, and let $\mathcal{E}$ be the event that the bound in Lemma 4 is violated for some $\boldsymbol{\Delta} \in \mathbb{B}_F(1)$. For $u, v = 1, 2, \cdots$, we define

$$\mathbb{S}_{u,v} := \left\{\boldsymbol{\Delta}\in\mathbb{B}_F(1) \mid 2^{u-1} \leq \sqrt{mn}\|\boldsymbol{\Delta}\|_\infty \leq 2^u, 2^{v-1} \leq \|\boldsymbol{\Delta}\|_* \leq 2^v\right\},$$
(42)

and let $\mathcal{E}_{u,v}$ be the event that the bound in Lemma 4 is violated for some $\boldsymbol{\Delta} \in \mathbb{S}_{u,v}$. We first show that

$$\mathcal{E} \subseteq \bigcup_{u,v=1}^{w}\mathcal{E}_{u,v}, \quad \text{where } w = \lceil\log_2 n\rceil.$$
(43)

For any matrix $\boldsymbol{\Delta} \in \mathbb{S}(\zeta,\theta)$, we have

$$\|\boldsymbol{\Delta}\|_* \geq \|\boldsymbol{\Delta}\|_F = 1 \text{ and } \|\boldsymbol{\Delta}\|_* \leq \sqrt{mn}\|\boldsymbol{\Delta}\|_F \leq n.$$

Similarly we have

$$n\|\boldsymbol{\Delta}\|_\infty \geq \sqrt{mn}\|\boldsymbol{\Delta}\|_\infty \geq 1 \text{ and } n\|\boldsymbol{\Delta}\|_\infty \leq n.$$

Then without loss of generality, we assume $\|\boldsymbol{\Delta}\|_* \in [1, n]$ and $n\|\boldsymbol{\Delta}\|_\infty \in [1, n]$. Therefore, if there exists a matrix $\boldsymbol{\Delta}$ of Frobenius norm one that violates the bound in Lemma 4, it must belong to some set $\mathbb{S}_{u,v}$ for some $u, v = 1, 2, \cdots, w$, where $w = \lceil\log_2 n\rceil$.

For $\zeta = 2^u$ and $\theta = 2^v$, we define the event

$$\tilde{\mathcal{E}}_{u,v} := \left\{Z(\zeta,\theta) \geq \frac{c_1'}{4}\zeta\theta\sqrt{\frac{n\log n}{|\Omega|}} + \frac{c_2'}{4}\zeta^2\frac{n\log n}{|\Omega|}\right\}.$$
(44)

If event $\mathcal{E}_{u,v}$ occurs, there must exist some $\boldsymbol{\Delta} \in \mathbb{S}_{u,v}$ such that

$$\left| \frac{mn}{|\Omega|} \|\mathcal{P}_\Omega(\boldsymbol{\Delta})\|_F^2 - 1 \right| \geq c_1' \sqrt{mn} \|\boldsymbol{\Delta}\|_\infty \|\boldsymbol{\Delta}\|_* \sqrt{\frac{n \log n}{|\Omega|}} + c_2' mn \|\boldsymbol{\Delta}\|_\infty^2 \frac{n \log n}{|\Omega|}$$

$$\geq c_1' 2^{u-1} 2^{v-1} \sqrt{\frac{n \log n}{|\Omega|}} + c_2' 2^{2(u-1)} \frac{n \log n}{|\Omega|} \tag{45}$$

$$\geq \frac{c_1'}{4} 2^u 2^v \sqrt{\frac{n \log n}{|\Omega|}} + \frac{c_2'}{4} (2^u)^2 \frac{n \log n}{|\Omega|}.$$

It means $\tilde{\mathcal{E}}_{u,v}$ occurs. Therefore, $\mathcal{E}_{u,v} \subseteq \tilde{\mathcal{E}}_{u,v}$.

Finally, we obtain

$$\mathbb{P}(\mathcal{E}) \leq \sum_{u,v=1}^{w} \mathbb{P}(\tilde{\mathcal{E}}_{u,v}) \leq w^2 e^{-n \log n} \leq e^{-\frac{1}{2} n \log n}, \tag{46}$$

where the first inequality holds due to the union bound and the third inequality holds due to $\log w^2 \leq \frac{1}{2} n \log n$. This finished the proof.

## 5.2 Proof for Lemma 6

*Proof.* For any matrix $\boldsymbol{\Delta} \in \mathbb{R}^{m \times n}$, denote its singular values by $\boldsymbol{\sigma} = [\sigma_1, \sigma_2, \cdots, \sigma_{\max\{m,n\}}]^T$. Define set $S = \{j \mid \sigma_j > \tau\}$. We have

$$\|\boldsymbol{\Delta}\|_* = \|\boldsymbol{\sigma}\|_1 = \|\boldsymbol{\sigma}_S\|_1 + \sum_{j \notin S} \sigma_j \leq \sqrt{|S|} \|\boldsymbol{\sigma}\|_2 + \tau \sum_{j \notin S} \frac{\sigma_j}{\tau}. \tag{47}$$

Since $\frac{\sigma_j}{\tau} \leq 1$ for all $j \notin S$, we obtain

$$\|\boldsymbol{\sigma}\|_1 \leq \sqrt{|S|} \|\boldsymbol{\sigma}\|_2 + \tau \sum_{j \notin S} \left(\frac{\sigma_j}{\tau}\right)^p$$

$$\leq \sqrt{|S|} \|\boldsymbol{\sigma}\|_2 + 2R_q \tau^{1-p} \tag{48}$$

Since $|S|\tau^p \leq \sum_{j \in S} \sigma_j^p \leq 2R_p$, we get

$$\|\boldsymbol{\sigma}\|_1 \leq \sqrt{2R_p} \tau^{-p/2} \|\boldsymbol{\sigma}\|_2 + 2R_p \tau^{1-P}. \tag{49}$$

Finally, we obtain $\|\boldsymbol{\Delta}\|_* \leq \sqrt{2R_p} \tau^{-p/2} \|\boldsymbol{\Delta}\|_F + 2R_p \tau^{1-p}$.

$\square$

## 5.3 Proof for Lemma 7

*Proof.* Denote $\boldsymbol{E}_{ij}$ the $(i,j)$ entry of $\boldsymbol{E}$.

$$\mathbb{P}(|\boldsymbol{E}_{ij}| \geq t\epsilon) \leq 2 \exp\left(-\frac{t^2}{2}\right). \tag{50}$$

Using Boole's inequality, we obtain

$$\mathbb{P}(\|\boldsymbol{E}\|_\infty \geq t\epsilon) \leq mn \mathbb{P}(|\boldsymbol{E}_{ij}| \geq t\epsilon) \leq 2mn \exp\left(-\frac{t^2}{2}\right) \leq 2n^2 \exp\left(-\frac{t^2}{2}\right). \tag{51}$$

It follows that

$$\mathbb{P}(\|\boldsymbol{E}\|_\infty \leq \epsilon\sqrt{2(c+2)\log n}) \geq 1 - 2n^{-c}. \tag{52}$$

Suppose $\{S_{ij}\}$ are independent Bernoulli$(|\Omega|/(mn))$ random variables. Denote $\boldsymbol{G}_{ij} = \boldsymbol{E}_{ij} S_{ij} \boldsymbol{e}_i \boldsymbol{e}_j^T$. Then $\mathbb{E}[\boldsymbol{G}_{ij}] = \boldsymbol{0}$, $\|\boldsymbol{G}_{ij}\|_2 \leq \|\boldsymbol{E}\|_\infty$, and $\mathcal{P}_\Omega(\boldsymbol{E}) = \sum_{ij} \boldsymbol{G}_{ij}$.

$$\|\sum_{ij} \mathbb{E}[\boldsymbol{G}_{ij} \boldsymbol{G}_{ij}^T]\|_2 = \|\sum_{ij} \mathbb{E}[\boldsymbol{E}_{ij}^2 S_{ij}^2 \boldsymbol{e}_i \boldsymbol{e}_i^T]\|_2$$

$$= \|\sum_{ij} \mathbb{E}[\boldsymbol{E}_{ij}^2] \mathbb{E}[S_{ij}] \mathbb{E}[\boldsymbol{e}_i \boldsymbol{e}_i^T]\|_2 \tag{53}$$

$$\leq \|\frac{\epsilon^2 |\Omega|}{mn} \sum_{ij} \mathbb{E}[\boldsymbol{e}_i \boldsymbol{e}_i^T]\|_2 = \frac{\epsilon^2 |\Omega|}{m}.$$

Similarly, we obtain

$$\| \sum_{ij} \mathbb{E} \left[ \boldsymbol{G}_{ij}^T \boldsymbol{G}_{ij} \right] \|_2 \leq \frac{\epsilon^2 |\Omega|}{n}. \tag{54}$$

As $m \leq n$, using matrix Bernstein, we have

$$\mathbb{P} \big( \| \sum_{ij} \boldsymbol{G}_{ij} \|_2 \geq t \big) \leq (m+n) \exp(-\frac{t^2}{2(\epsilon^2 |\Omega| m^{-1} + t\epsilon \sqrt{2(c+2)\log n}/3)}). \tag{55}$$

Suppose $\epsilon^2 |\Omega| m^{-1} \geq t\epsilon \sqrt{2(c+2)\log n}/3$, namely, $t \leq \dfrac{\epsilon |\Omega|}{m\sqrt{2(c+2)\log n}/3}$, inequality (55) becomes

$$\mathbb{P} \big( \| \sum_{ij} \boldsymbol{G}_{ij} \|_2 \geq t \big) \leq (m+n) \exp(-\frac{t^2}{4\epsilon^2 |\Omega| m^{-1}}). \tag{56}$$

Then we have

$$\mathbb{P} \big( \| \sum_{ij} \boldsymbol{G}_{ij} \|_2 \leq 2\epsilon \sqrt{\frac{c'|\Omega|\log n}{m}} \big) \geq 1 - \frac{2}{n^{c'-1}}. \tag{57}$$

Because we have assumed that $t \leq \dfrac{\epsilon |\Omega|}{m\sqrt{2(c+2)\log n}/3}$, the above inequality holds when $|\Omega| \geq \frac{8}{9}c'(c+2)n\log^2 n$. Letting $c' = 3$ and $c = 2$ and considering the probability in (52) and (57), the following inequality holds with probability at least $1 - 4n^{-2}$

$$\| \mathcal{P}_\Omega(\boldsymbol{E}) \|_2 \leq 2\sqrt{3}\epsilon \sqrt{\frac{|\Omega|\log n}{m}}, \tag{58}$$

provided that $|\Omega| \geq \frac{32}{3} n \log^2 n$.

$\square$

# 6 Optimization for low-rank matrix completion

In this section, first, we detail the ADMM [5, 6, 7] (with linearization) optimization for the following matrix completion problem:

$$\begin{aligned} \underset{\boldsymbol{X},\boldsymbol{A},\boldsymbol{B},\boldsymbol{E}}{\text{minimize}} \ &\|\boldsymbol{A}\|_{2,1} + \frac{\alpha}{2}\|\boldsymbol{B}\|_F^2 + \frac{\beta}{2}\|\boldsymbol{E}\|_F^2, \\ \text{subject to} \ &\boldsymbol{X} = \boldsymbol{A}\boldsymbol{B} + \boldsymbol{E}, \ P_\Omega(\boldsymbol{X}) = P_\Omega(\boldsymbol{M}_e). \end{aligned} \tag{59}$$

We construct the augmented Lagrange function

$$\mathcal{L}(\boldsymbol{X},\boldsymbol{A},\boldsymbol{B},\boldsymbol{E},\boldsymbol{Y}) \triangleq \|\boldsymbol{A}\|_{2,1} + \frac{\alpha}{2}\|\boldsymbol{B}\|_F^2 + \frac{\beta}{2}\|\boldsymbol{E}\|_F^2 + \langle \boldsymbol{X} - \boldsymbol{A}\boldsymbol{B} - \boldsymbol{E}, \boldsymbol{Y} \rangle + \frac{\mu}{2}\|\boldsymbol{X} - \boldsymbol{A}\boldsymbol{B} - \boldsymbol{E}\|_F^2, \tag{60}$$

where $\boldsymbol{Y} \in \mathbb{R}^{m \times n}$ denotes the multipliers and $\mu$ is a parameter. Here we have implicitly include the constraint $P_\Omega(\boldsymbol{X}) = P_\Omega(\boldsymbol{M}_e)$ in $\mathcal{L}$ by requiring $\boldsymbol{X}$ satisfies $P_\Omega(\boldsymbol{X}) = P_\Omega(\boldsymbol{M}_e)$. The alternate updating steps are

$$\begin{cases} \boldsymbol{X}_t = \underset{P_\Omega(\boldsymbol{X})=P_\Omega(\boldsymbol{M}_e)}{\text{argmin}} \ \mathcal{L}(\boldsymbol{X}, \boldsymbol{A}_{t-1}, \boldsymbol{B}_{t-1}, \boldsymbol{E}_{t-1}, \boldsymbol{Y}_{t-1}) \\ \boldsymbol{A}_t = \underset{\boldsymbol{A}}{\text{argmin}} \ \hat{\mathcal{L}}(\boldsymbol{X}_t, \boldsymbol{A}, \boldsymbol{B}_{t-1}, \boldsymbol{E}_{t-1}, \boldsymbol{Y}_{t-1}) \\ \boldsymbol{B}_t = \underset{\boldsymbol{B}}{\text{argmin}} \ \mathcal{L}(\boldsymbol{X}_t, \boldsymbol{A}_t, \boldsymbol{B}, \boldsymbol{E}_{t-1}, \boldsymbol{Y}_{t-1}) \\ \boldsymbol{E}_t = \underset{\boldsymbol{E}}{\text{argmin}} \ \mathcal{L}(\boldsymbol{X}_t, \boldsymbol{A}_t, \boldsymbol{B}_t, \boldsymbol{E}, \boldsymbol{Y}_{t-1}) \\ \boldsymbol{Y}_t = \boldsymbol{Y}_{t-1} + \mu(\boldsymbol{X}_t - \boldsymbol{A}_t\boldsymbol{B}_t - \boldsymbol{E}_t), \end{cases} \tag{61}$$

where $t$ denotes the iteration number. In (61), $\hat{\mathcal{L}}$ denotes the linearization for $\mathcal{L}$ and will be detailed later. We now explain how to update $\boldsymbol{X}$, $\boldsymbol{A}$, $\boldsymbol{B}$, and $\boldsymbol{E}$ in the first four lines of (61). Specifically, we update $\boldsymbol{X}$ via solving

$$\boldsymbol{X}_t = \operatorname*{argmin}_{P_\Omega(\boldsymbol{X})=P_\Omega(\boldsymbol{M}_e)} \frac{\mu}{2}\|\boldsymbol{X} - \boldsymbol{A}_{t-1}\boldsymbol{B}_{t-1} - \boldsymbol{E}_{t-1} + \mu^{-1}\boldsymbol{Y}_{t-1}\|_F^2. \tag{62}$$

The solution is

$$\boldsymbol{X}_t = P_{\bar{\Omega}}(\boldsymbol{A}_{t-1}\boldsymbol{B}_{t-1} + \boldsymbol{E}_{t-1} - \mu^{-1}\boldsymbol{Y}_{t-1}) + P_\Omega(\boldsymbol{M}_e), \tag{63}$$

where $\bar{\Omega}$ denotes the locations of unknown entries. We update $\boldsymbol{A}$ via solving

$$\boldsymbol{A}_t = \operatorname*{argmin}_{\boldsymbol{A}} \|\boldsymbol{A}\|_{2,1} + \frac{\mu}{2}\|\boldsymbol{X}_t - \boldsymbol{A}\boldsymbol{B}_{t-1} - \boldsymbol{E}_{t-1} + \mu^{-1}\boldsymbol{Y}_{t-1}\|_F^2, \tag{64}$$

which, however, has no closed-form solution. We can linearize $\frac{\mu}{2}\|\boldsymbol{X}_t - \boldsymbol{A}\boldsymbol{B}_{t-1} - \boldsymbol{E}_{t-1} + \mu^{-1}\boldsymbol{Y}_{t-1}\|_F^2$ at $\boldsymbol{A}_{t-1}$ and have

$$\begin{aligned}
\boldsymbol{A}_t &= \operatorname*{argmin}_{\boldsymbol{A}} \|\boldsymbol{A}\|_{2,1} + \frac{\mu}{2}\|\boldsymbol{X}_{t-1} - \boldsymbol{A}_{t-1}\boldsymbol{B}_{t-1} - \boldsymbol{E}_{t-1} + \mu^{-1}\boldsymbol{Y}_{t-1}\|_F^2 \\
&\quad + \langle \boldsymbol{Q}, \boldsymbol{A} - \boldsymbol{A}_{t-1}\rangle + \frac{L_t}{2}\|\boldsymbol{A} - \boldsymbol{A}_{t-1}\|_F^2 \\
&= \operatorname*{argmin}_{\boldsymbol{A}} \|\boldsymbol{A}\|_{2,1} + \frac{L_t}{2}\|\boldsymbol{A} - \boldsymbol{A}_{t-1} + L_t^{-1}\boldsymbol{Q}\|_F^2,
\end{aligned} \tag{65}$$

where $\boldsymbol{Q} = \mu(\boldsymbol{X}_t - \boldsymbol{A}_{t-1}\boldsymbol{B}_{t-1} - \boldsymbol{E}_{t-1} + \mu^{-1}\boldsymbol{Y}_{t-1})(-\boldsymbol{B}_{t-1}^T)$ and $L_t \geq \mu\|\boldsymbol{B}_{t-1}\|_2^2$. The closed-form solution of (65) is

$$\boldsymbol{A}_t = \Phi_{L_t^{-1}}(\boldsymbol{A}_{t-1} - L_t^{-1}\boldsymbol{Q}), \tag{66}$$

where $\Phi_\tau(\cdot)$ is the column-wise soft-thresholding [8] operator defined as

$$\Phi_\tau(\boldsymbol{u}) = \begin{cases} \dfrac{(\|\boldsymbol{u}\| - \tau)\boldsymbol{u}}{\|\boldsymbol{u}\|}, & \text{if } \|\boldsymbol{u}\| > \tau; \\ \boldsymbol{0}, & \text{otherwise.} \end{cases} \tag{67}$$

We then update $\boldsymbol{B}$ as

$$\begin{aligned}
\boldsymbol{B}_t &= \operatorname*{argmin}_{\boldsymbol{B}} \frac{\alpha}{2}\|\boldsymbol{B}\|_F^2 + \frac{\mu}{2}\|\boldsymbol{X}_t - \boldsymbol{A}_t\boldsymbol{B} - \boldsymbol{E}_{t-1} + \mu^{-1}\boldsymbol{Y}_{t-1}\|_F^2 \\
&= (\mu\boldsymbol{A}_t^T\boldsymbol{A}_t + \alpha\boldsymbol{I}_d)^{-1}(\boldsymbol{A}_t^T(\boldsymbol{X}_t - \boldsymbol{E}_{t-1} + \mu^{-1}\boldsymbol{Y}_{t-1})).
\end{aligned} \tag{68}$$

It is easy to show that, when one column of $\boldsymbol{A}_t$ is zero, the corresponding row of $\boldsymbol{B}_t$ given by (68) is also zero. Finally, we update $\boldsymbol{E}$ as

$$\begin{aligned}
\boldsymbol{E}_t &= \operatorname*{argmin}_{\boldsymbol{E}} \frac{\beta}{2}\|\boldsymbol{E}\|_F^2 + \frac{\mu}{2}\|\boldsymbol{X}_t - \boldsymbol{A}_t\boldsymbol{B}_t - \boldsymbol{E} + \mu^{-1}\boldsymbol{Y}_{t-1}\|_F^2 \\
&= \frac{\mu}{\beta + \mu}(\boldsymbol{X}_t - \boldsymbol{A}_t\boldsymbol{B}_t + \mu^{-1}\boldsymbol{Y}_{t-1}).
\end{aligned} \tag{69}$$

The optimization steps are summarized in Algorithm 1.

Consider the general problem

$$\operatorname*{minimize}_{\boldsymbol{A},\boldsymbol{B}} \frac{1}{2}\|\mathcal{P}_\Omega(\boldsymbol{M}_e - \boldsymbol{A}\boldsymbol{B})\|_F^2 + \gamma\big(\|\boldsymbol{A}\|_{2,q}^q + \frac{\alpha}{2}\|\boldsymbol{B}^T\|_F^2\big), \tag{70}$$

where $q = 1, \frac{1}{2}, \frac{1}{4}, \cdots$. When $q = 1$, the corresponding PALM optimization is shown in Algorithm 2, in which $\boldsymbol{C}$ is a binary matrix with 1 and 0 corresponding to the observed and missing entries of $\boldsymbol{M}_e$ respectively.

When $q \neq 1$, we propose to solve (70) via PALM coupled with iteratively reweighted minimization [9]. For instance, the subproblem of the term $\boldsymbol{A}$ during the iterations of PALM is of the form

$$\operatorname*{minimize}_{\boldsymbol{A}} f(\boldsymbol{A}) + \|\boldsymbol{A}\|_{2,q}^q, \tag{71}$$

**Algorithm 1** FGSR$_{2/3}$ for LRMC (59) solved by ADMM

---

**Input:** $\boldsymbol{M}_e, \Omega, d \geq r, \alpha, \beta, \mu, t_{\max}, t = 0, \boldsymbol{Y}_0 = \boldsymbol{E}_0 = \boldsymbol{0}, \boldsymbol{X}_0, \boldsymbol{A}_0, \boldsymbol{B}_0$
 1: **repeat**
 2:     $t \leftarrow t + 1$
 3:     $\boldsymbol{X}_t = P_{\bar{\Omega}}(\boldsymbol{A}_{t-1}\boldsymbol{B}_{t-1} + \boldsymbol{E}_{t-1} - \mu^{-1}\boldsymbol{Y}_{t-1}) + P_{\Omega}(\boldsymbol{M}_e)$
 4:     $\boldsymbol{Q} = \mu(\boldsymbol{X}_t - \boldsymbol{A}_{t-1}\boldsymbol{B}_{t-1} - \boldsymbol{E}_{t-1} + \mu^{-1}\boldsymbol{Y}_{t-1})(-\boldsymbol{B}_{t-1}^T)$
 5:     $L_t = 1.01\mu\|\boldsymbol{B}_{t-1}\boldsymbol{B}_{t-1}^T\|_2$
 6:     $\boldsymbol{A}_t = \Phi_{L_t^{-1}}(\boldsymbol{A}_{t-1} - L_t^{-1}\boldsymbol{Q})$
 7:     $\boldsymbol{B}_t = (\mu\boldsymbol{A}_t^T\boldsymbol{A}_t + \alpha\boldsymbol{I}_d)^{-1}(\boldsymbol{A}_t^T(\boldsymbol{X}_t - \boldsymbol{E}_{t-1} + \frac{1}{\mu}\boldsymbol{Y}_{t-1}))$
 8:     $\boldsymbol{E}_t = \dfrac{\mu}{\beta + \mu}(\boldsymbol{X}_t - \boldsymbol{A}_t\boldsymbol{B}_t + \mu^{-1}\boldsymbol{Y}_{t-1})$
 9:     $\boldsymbol{Y}_t = \boldsymbol{Y}_{t-1} + \mu(\boldsymbol{X}_t - \boldsymbol{A}_t\boldsymbol{B}_t - \boldsymbol{E}_t)$
10:     $d \leftarrow \text{nnzc}(\boldsymbol{A}_t)$
11:     Remove the zero columns of $\boldsymbol{A}_t$ and $\boldsymbol{B}_t^T$
12: **until** converged or $t = t_{\max}$
**Output:** $\boldsymbol{M} = \boldsymbol{X}_t$

---

**Algorithm 2** FGSR$_{2/3}$ for LRMC (70) ($q = 1$) solved by PALM

---

**Input:** $\boldsymbol{M}_e, \Omega, d \geq r, \alpha, \gamma, 0.1 \leq \eta \leq 1, t_{\max}, t = 0, \boldsymbol{A}_0, \boldsymbol{B}_0$
 1: **repeat**
 2:     $t \leftarrow t + 1$
 3:     $\boldsymbol{Q} = \big(\boldsymbol{C} \odot (\boldsymbol{M}_e - \boldsymbol{A}_{t-1}\boldsymbol{B}_{t-1})\big)(-\boldsymbol{B}_{t-1}^T)$
 4:     $L_t = 1.01\eta\|\boldsymbol{B}_{t-1}\boldsymbol{B}_{t-1}^T\|_2$
 5:     $\boldsymbol{A}_t = \Phi_{\gamma/L_t}(\boldsymbol{A}_{t-1} - L_t^{-1}\boldsymbol{Q})$
 6:     $L_t = 1.01\eta\|\boldsymbol{A}_t^T\boldsymbol{A}_t\|_2$
 7:     $\boldsymbol{B}_t = \frac{1}{\alpha\gamma+L_t}\big(\boldsymbol{A}_t^T(\boldsymbol{C} \odot (\boldsymbol{M}_e - \boldsymbol{A}_t\boldsymbol{B}_{t-1})) + L_t\boldsymbol{B}_{t-1}\big)$;
 8:     $d \leftarrow \text{nnzc}(\boldsymbol{A}_t)$
 9:     Remove the zero columns of $\boldsymbol{A}_t$ and $\boldsymbol{B}_t^T$
10: **until** converged or $t = t_{\max}$
**Output:** $\boldsymbol{M} = \boldsymbol{A}_t\boldsymbol{B}_t$

---

for some function $f$. At iteration $t$, we solve

$$\underset{\boldsymbol{A}}{\text{minimize}}\, f(\boldsymbol{A}) + \sum_j w_j\|\boldsymbol{A}_{:j}\|_2^2, \tag{72}$$

where $w_j = (\|\boldsymbol{A}_{:j}^{(t-1)}\|_2 + \xi)^{q-2}$, $\xi$ is a small number, and $\boldsymbol{A}^{(t-1)}$ denotes the $\boldsymbol{A}$ obtained at iteration $t - 1$.

In addition, when we use FGSR$_{1/2}$, the optimizations are similar to those of (59) and (70).

# 7 Optimization for robust PCA

In this section, we first detail the ADMM with linearization for the following FGSR$_{2/3}$ based RPCA problem:

$$\begin{aligned}
&\underset{\boldsymbol{A}, \boldsymbol{B}, \boldsymbol{E}}{\text{minimize}}\, \|\boldsymbol{A}\|_{2,1} + \frac{\alpha}{2}\|\boldsymbol{B}\|_F^2 + \lambda\|\boldsymbol{E}\|_1, \\
&\text{subject to } \boldsymbol{M}_e = \boldsymbol{A}\boldsymbol{B} + \boldsymbol{E}.
\end{aligned} \tag{73}$$

Here $\lambda$ is a regularization parameter and $\|\cdot\|_1$ is the $\ell_1$ norm of matrix enforcing sparsity. We minimize the augmented Lagrange function alternately

$$\begin{aligned}
\mathcal{L}(\boldsymbol{E}, \boldsymbol{A}, \boldsymbol{B}, \boldsymbol{Y}) &:= \|\boldsymbol{A}\|_{2,1} + \frac{\alpha}{2}\|\boldsymbol{B}\|_F^2 + \lambda\|\boldsymbol{E}\|_1 \\
&+ \langle \boldsymbol{M}_e - \boldsymbol{A}\boldsymbol{B} - \boldsymbol{E}, \boldsymbol{Y}\rangle + \frac{\mu}{2}\|\boldsymbol{M}_e - \boldsymbol{A}\boldsymbol{B} - \boldsymbol{E}\|_F^2,
\end{aligned} \tag{74}$$

The optimization of RPCA is similar to that of LRMC detailed in the previous section. One major difference is the presence of $\boldsymbol{E}$, which results in the following subproblem

$$\underset{\boldsymbol{E}}{\text{minimize}} \; \lambda\|\boldsymbol{E}\|_1 + \frac{\mu}{2}\|\boldsymbol{M}_e - \boldsymbol{A}\boldsymbol{B} - \boldsymbol{E} + \mu^{-1}\boldsymbol{Z}\|_F^2. \tag{75}$$

The solution of (75) is

$$\boldsymbol{E} = \Psi_{\lambda/\mu}(\boldsymbol{M}_e - \boldsymbol{A}\boldsymbol{B} + \mu^{-1}\boldsymbol{Z}), \tag{76}$$

where $\Psi_\tau(\cdot)$ is the element-wise soft-thresholding operator [8] defined as

$$\Psi_\tau(u) = \frac{u}{|u|}\max(|u| - \tau, 0). \tag{77}$$

The optimization is shown in Algorithm 3.

---

**Algorithm 3** FGSR$_{2/3}$ for RPCA solved by ADMM

---

**Input:** $\boldsymbol{M}_e, \Omega, d \geq r, \alpha, \lambda, \mu, t_{\max}, t = 0, \boldsymbol{Y}_0 = \boldsymbol{0}, \boldsymbol{E}_0 = \boldsymbol{0}, \boldsymbol{A}_0, \boldsymbol{B}_0$
1: **repeat**
2:     $t \leftarrow t + 1$
3:     $\boldsymbol{E}_t = \Psi_{\lambda/\mu}(\boldsymbol{M}_e - \boldsymbol{A}_{t-1}\boldsymbol{B}_{t-1} + \mu^{-1}\boldsymbol{Y}_{t-1})$
4:     $\boldsymbol{Q} = \mu(\boldsymbol{M}_e - \boldsymbol{A}_{t-1}\boldsymbol{B}_{t-1} - \boldsymbol{E}_t + \mu^{-1}\boldsymbol{Y}_{t-1})(-\boldsymbol{B}_{t-1}^T)$
5:     $L_t = 1.01\mu\|\boldsymbol{B}_{t-1}\boldsymbol{B}_{t-1}^T\|_2$
6:     $\boldsymbol{A}_t = \Phi_{L_t^{-1}}(\boldsymbol{A}_{t-1} - L_t^{-1}\boldsymbol{Q})$
7:     $\boldsymbol{B}_t = (\mu\boldsymbol{A}_t^T\boldsymbol{A}_t + \alpha\boldsymbol{I}_d)^{-1}(\boldsymbol{A}_t^T(\boldsymbol{M}_e - \boldsymbol{E}_t + \frac{1}{\mu}\boldsymbol{Y}_{t-1}))$
8:     $\boldsymbol{Y}_t = \boldsymbol{Y}_{t-1} + \mu(\boldsymbol{M}_e - \boldsymbol{A}_t\boldsymbol{B}_t - \boldsymbol{E}_t)$
9:     $d \leftarrow \text{nnzc}(\boldsymbol{A}_t)$
10:    Remove the zero columns of $\boldsymbol{A}_t$ and $\boldsymbol{B}_t^T$
11: **until** converged or $t = t_{\max}$
**Output:** $\boldsymbol{M} = \boldsymbol{A}_t\boldsymbol{B}_t, \boldsymbol{E} = \boldsymbol{E}_t$

---

Consider the following model

$$\boldsymbol{M}_{ef} = \boldsymbol{M} + \boldsymbol{E} + \boldsymbol{F}, \tag{78}$$

where $\boldsymbol{F}$ denotes small Gaussian noises. We then have the following general problem

$$\underset{\boldsymbol{A},\boldsymbol{B},\boldsymbol{E}}{\text{minimize}} \; \|\boldsymbol{A}\|_{2,q}^q + \frac{\alpha}{2}\|\boldsymbol{B}\|_F^2 + \lambda\|\boldsymbol{E}\|_1 + \frac{\beta}{2}\|\boldsymbol{A}\boldsymbol{B} + \boldsymbol{E} - \boldsymbol{M}_{ef}\|_F^2, \tag{79}$$

where $q \in \{1, \frac{1}{2}, \frac{1}{4}, \cdots\}$. The optimization can be solved via PALM plus iteratively reweighted minimization when $q \neq 1$, shown in (72). In addition, the optimization of FGSR$_{1/2}$ based RPCA are similar to those of (73) and (79). Empirically, we found that $\lambda = (\max(m, n))^{-0.5/p}$ works well for (73) and (79), where $p$ is the $p$ in the Schatten-$p$ norm induced by FGSR.

# 8 On the convergence

The convergence of PALM for general nonconvex and nonsmooth problems have been analyzed in [10, 11, 12]. The convergence of ADMM for general nonconvex and nonsmooth problems [6] gains increasing attention in recent years. For example, Wang et al. [5] proved the convergence of ADMM on a class of nonconvex problems with linear constraints. Gao et al. [7] proved the convergence of nonconvex ADMM with multiaffine (nonlinear and nonconvex) constraints. However, the convergence of ADMM for solving (59) with small $\|\boldsymbol{E}\|_F^2$ and (73) is still an open problem [13]. The high difficulty in convergence analysis arises from that the dual variable is associated with a separable primary variable (e.g. $\boldsymbol{E}$ in (73)) that has a nonsmooth function or the dual variable is associated with two nonlinearly coupled primary variables (e.g. $\boldsymbol{A}$ and $\boldsymbol{B}$ in (59)) even if their functions are strongly convex and have Lipschitz-continuous gradients. Proving the convergence of ADMM for these cases is out of the scope of our paper. In practice, even if without theoretical convergence guarantee, ADMM for (59) and (73) often converge to a stationary point (or even global minimum), provided that $\mu$ is not too small.

# 9 More about the numerical results

All experiments are conducted on a computer with Inter-i7-3.4GHz Core and 16GB RAM. All results we report are the average of ten repeated trials.

## 9.1 Matrix completion

**Optimization, codes, and hyper-parameters** For the nuclear norm minimization (solved by inexact augmented Lagrange multiplier method) [14], truncated nuclear norm minimization [17], and Riemannian pursuit [18], we utilize the codes provided by the authors of the corresponding papers. We implement the weighted nuclear norm minimization [19], Schatten-$p$ norm minimization (solved by iteratively reweighted minimization [9]), max norm minimization (solved by alternating projected gradient method) [15], Bi-Nuclear norm minimization [16], and $F^2$+nuclear norm minimization [20] via MATLAB according to the algorithms described in the corresponding papers.

In noiseless matrix completion, we solve the minimization of F-nuclear norm, FGSR$_{2/3}$ ($\alpha = 1$), and FGSR$_{1/2}$ by ADMM with linearization. The Lagrange parameter $\mu$ in all related methods is selected from $\{0.0001, 0.001, 0.01, 0.1, 1\}$.

In noisy matrix completion, the minimizations of F-nuclear norm, FGSR$_{2/3}$ ($\alpha = 1$), and FGSR$_{1/2}$ by PALM. In all methods, the hyper-parameters (e.g. $\gamma^{-1}$ in (70)) corresponding to the noise term are carefully determined to provide the best performances. In all cases excluding those studying the effect of rank initialization, we set $d = 1.5r$ for F-nuclear norm, max norm, and FGSR.

**Clean synthetic data** We generate low-rank matrices via

$$\boldsymbol{M} = \boldsymbol{LWR}^T, \tag{80}$$

where the entries of $\boldsymbol{L} \in \mathbb{R}^{m \times r}$ and $\boldsymbol{R} \in \mathbb{R}^{n \times r}$ are randomly drawn from $\mathcal{N}(0, 1)$. $\boldsymbol{W} = \mathrm{diag}([1 + 0.1(r-1), 1+0.1(r-2), \cdots, 1.1, 1])$, which simulates that real data matrices often have degenerated singular values. In this paper, we set $m = n = 500$. The performance of matrix completion is evaluated by

$$\text{Relative recovery error} := \frac{\|\mathcal{P}_{\bar{\Omega}}(\boldsymbol{M} - \hat{\boldsymbol{M}})\|_F}{\|\mathcal{P}_{\bar{\Omega}}(\boldsymbol{M})\|_F}, \tag{81}$$

where $\hat{\boldsymbol{M}}$ denotes the recovered matrix and $\bar{\Omega}$ denotes the locations of missing entries.

Figure 1 shows the recovery error of some methods in the cases of different rank and different missing rate. Our FGSR$_{2/3}$ and FGSR$_{1/2}$ outperformed other methods when the rank or/and missing rate are relatively high.

Figure 1: Matrix completion on clean synthetic data: the relative recovery error in the cases of different rank and different missing rate

**Noisy synthetic data** Suppose the observed entries of $\boldsymbol{M}$ are corrupted by Gaussian noise, i.e. $[\boldsymbol{M}_e]_{ij} = \boldsymbol{M}_{ij} + e_{ij}$, $(i, j) \in \Omega$, where $\boldsymbol{M}$ is given by (80) and $e_{ij} \sim \mathcal{N}(0, \epsilon^2)$. Then the signal noise ratio is SNR $= \sigma/\sigma_e$, where $\sigma$ denotes the standard deviation of the entries of $\boldsymbol{M}$.

**Real data** The MovieLens-1M dataset[1] consists of 1 million ratings ($1 \sim 5$) for 3900 movies by 6040 users. The movies rated by less than 5 users are deleted in this study because the corresponding

ratings may never be recovered when the matrix rank is higher than 5. Then the size of the processed matrix is $3416 \times 6040$. Since the original matrix is highly incomplete, we randomly sample 70% or 50% of the known ratings of each user and perform matrix completion. The performance is evaluated by the normalized mean absolute error (NMAE) [21, 22] and normalized root-mean-squared-error (RMSE) [22]. They are defined by

$$\text{NMAE} := \frac{1}{(\boldsymbol{M}_{\max} - \boldsymbol{M}_{\min})|\Upsilon \setminus \Omega|} \sum_{(i,j) \in \Upsilon \setminus \Omega} |\boldsymbol{M}_{ij} - \hat{\boldsymbol{M}}_{ij}|, \tag{82}$$

and

$$\text{RMSE} := \sqrt{\frac{\sum_{(i,j) \in \Upsilon \setminus \Omega}(\boldsymbol{X}_{ij} - \hat{\boldsymbol{X}}_{ij})^2}{\sum_{(i,j) \in \Upsilon \setminus \Omega} \boldsymbol{X}_{ij}^2}}, \tag{83}$$

where $\Upsilon$ denotes the entries known originally and $\Omega$ denotes the entries we sampled from $\Upsilon$.

## 9.2 RPCA

**Synthetic data** The corrupted matrix is $\boldsymbol{M}_e = \boldsymbol{M} + \boldsymbol{E}$, where $\boldsymbol{M}$ is generated by (80) and $\boldsymbol{E}$ is a sparse matrix. The nonzero entries of $\boldsymbol{E}$ are drawn from $\mathcal{N}(0, \epsilon^2)$. Then the signal noise ratio in terms of the corrupted entries is $\text{SNR}_c := \sigma/\sigma_e$, where $\sigma$ denotes the standard deviation of the entries of $\boldsymbol{M}$. We call the fraction of nonzero entries of $\boldsymbol{E}$ noise density. The performance of RPCA is evaluated by

$$\text{Relative recovery error} := \frac{\|\boldsymbol{M} - \hat{\boldsymbol{M}}\|_F}{\|\boldsymbol{M}\|_F}, \tag{84}$$

where $\hat{\boldsymbol{M}}$ denotes the recovered matrix.

Figure 2 shows the performance of nuclear norm, F-nuclear norm, FGSR$_{2/3}$, and FGSR$_{1/2}$ in the cases of different noise density and different rank. FGSR$_{2/3}$ and FGSR$_{1/2}$ outperformed nuclear norm and F-nuclear norm.

Figure 2: RPCA on synthetic data: the relative recovery error in the case of different rank and noise density ($\text{SNR}_c = 1$).

**Natural images** The pixel matrices[2] of many natural images are of low-rank approximately, which enables us to use RPCA to remove sparse noises in the images. We consider two images of size $796 \times 834$ and $768 \times 1024$ respectively. We reconstructed each image via preserving the largest 50 singular values to form exactly low-rank matrices in the 3 color channels. Then we added salt-and-pepper noise of density 40% to the two images and perform RPCA in each color channel individually. The clean images, corrupted images, and recovery results are shown in Figure 3 and Figure 4. Both numerically and visually, the performance of our FGSR$_{2/3}$ is better than those of nuclear norm and F-nuclear norm. The results of FGSR$_{1/2}$ are nearly the sane as those of FGSR$_{2/3}$ and are omitted in this paper.

Figure 3: RPCA in image denoising (example 1)

Figure 4: RPCA in image denoising (example 2)

## Footnotes

[1]https://grouplens.org/datasets/movielens/

[2]If the pixel matrix of an image can not be well approximated by a low-rank matrix, we can extract small patches of the image as vectors to form a matrix (approximately low-rank) and then perform RPCA.