[Reviews · NeurIPS 2019]

Reviewer 1



This study links "factor group-sparse regularized matrix factorization" and "Schatten-p-norm regularization with p=1/2 or 2/3" theoretically. This perspective is interesting. However, the proposed method and results do not have big impact in practical perspective because the convex regularized matrix factorization itself is very naive, and non-convex regularized low-rank matrix recovery is now widely studied and there are many related works such as truncated nuclear-norm[ex1], weighted nuclear-norm[ex2], capped-l1[ex3], LSP[ex4], SCAD[ex5], and MCP[ex6]. [ex1] Y. Hu, D. Zhang, J. Ye, X. Li, and X. He, “Fast and accurate matrix completion via truncated nuclear norm regularization,” IEEE Transactions on Pattern Analysis and Machine Intelligence, vol. 35, no. 9, pp. 2117–2130, 2013. [ex2] Gu, Shuhang, et al. "Weighted nuclear norm minimization with application to image denoising." Proceedings of the IEEE conference on computer vision and pattern recognition. 2014. [ex3] T. Zhang, “Analysis of multi-stage convex relaxation for sparse regularization,” Journal of Machine Learning Research, vol. 11, pp. 1081–1107, 2010. [ex4] E. Candes, M. Wakin, and S. Boyd, “Enhancing sparsity by reweighted l1 minimization,” Journal of Fourier Analysis and Applications, vol. 14, no. 5-6, pp. 877–905, 2008. [ex5] J. Fan and R. Li, “Variable selection via nonconcave penalized likelihood and its oracle properties,” Journal of the American Statistical Association, vol. 96, no. 456, pp. 1348–1360, 2001. [ex6] C. Zhang, “Nearly unbiased variable selection under minimax concave penalty,” Annals of Statistics, vol. 38, no. 2, pp. 894–942, 2010. Also in another perspective, greedy rank-increment approach [ex7,ex8,ex9] for low-rank matrix recovery should be referred for discussion. This does not need to estimate initial d unlike regularized matrix factorization methods, and it is usually memory efficient. [ex7] M. Tan, I. W. Tsang, L. Wang, B. Vandereycken, and S. J. Pan, “Riemannian pursuit for big matrix recovery,” in Proc. 31st Int. Conf. Mach. Learn. (ICML), pp. 1539–1547, 2014. [ex8] Yokota, Tatsuya, and Andrzej Cichocki. "A fast automatic low-rank determination algorithm for noisy matrix completion." 2015 Asia-Pacific Signal and Information Processing Association Annual Summit and Conference (APSIPA). IEEE, 2015. [ex9] A. Uschmajew and B. Vandereycken, “Greedy rank updates combined with Riemannian descent methods for low-rank optimization,” in Proc. Int. Conf. Sampl. Theory Appl. (SampTA), pp. 420–424, 2015.

Reviewer 2



This paper proposed a new non-convex regularizer for the low-rank matrix analysis, which can get rid of SVD, and apply to large-scale matrix analysis. The theoretical analysis gives the bounds for the proposed method.

Reviewer 3



This paper focuses on the problem of the low-rank matrix recovery, and proposes a factor group-sparse regularizers as a relaxation of the number of nonzero columns in a factorization of the matrix. It seek to minimize the number of nonzero columns of the factored matrices. Analysis is conducted. However, there are some concerns to be addressed. -The motivation is somewhat unclear. In introduction, some contents should be presented in the related work section or the Preliminary section to well show the motivation of this work. -My another concern is about the value of p. In real application, how to set the suitable values of p? Is there any suggestion? -In experiments, how about the results of FGSR-1/2? It is better to show the results of different values of p.

Reviewer 4



Updated after author feedback: The authors did not have the opportunity to address my concerns about the experimental setup, as this review was solicited after the response period. But I think the changes they have made in response to other reviewers (comparisons with non-convex estimators) have increased the persuasiveness of their experimental comparisons. My score remains 7. The authors extend the factorization of the nuclear norm into the sum of squared Frobenius norms to find factorized expressions for the low-rank promoting Shatten quasi-norms. The advantage of these factorized expressions is that they are weighted powers of the column norms of the factorizations, so are computationally attractive in one sense. On the other hand, they are non-convex. The authors demonstrate empirically that these estimators perform well in practice using their suggested optimization algorithms, outperforming prior factorization and nuclear norm-based formulations of low rank matrix recovery problems. The paper is well-written, with the exception of the experimental section which is too vague about how the experiments were conducted. The results should be averaged over multiple runs and standard deviations should be described. The procedure for choosing the hyperparameters of the methods should be described so the readers can be sure that the comparisons are fair. How were the methods implemented (especially, what algorithms were used for the baseline methods?). I am concerned by the fact that in Figure 1d the factorized nuclear norm approach is slower than the FGSR-2/3 method: they are both sums of weighted powers of column norms of the factorizations, but the former is a smooth objective, while the latter is not. What is causing this counter-intuitive result? Overall, the main contribution (the factorized representation for the Shatten quasi-norms) is interesting and useful in itself, and the resulting algorithms are reasonable, but the experimental evaluation is questionable. Thus my score is 7: I recommend accept, although I will be fine with a reject.

[Author Response · NeurIPS 2019]

We answered all questions posed by the reviewers and added comparisons with other five algorithms they asked about.
Our methods still provide lower recovery error than any competitor. We sincerely thank the reviewers for their comments
and time spent on our paper.

**Response to reviewer** #1    We propose to estimate $d$ as $d = |\Omega|/(m+n)$. Given a rank-$r$ matrix $X \in \mathbb{R}^{m \times n}$, the
number of degrees of freedom is $(m+n)r - r^2$. Suppose the number of observed entries is $|\Omega|$. Then $|\Omega| \geq (m+n)r - r^2$
((1)) should hold; otherwise, $X$ can not be determined uniquely. Considering incoherence property and random sampling,
Candès and Recht (2009) proved that the minimum number of observed entries required to recovery $X$ (whatever
methods used) with high probability is $C\mu n r \log n$ (suppose $m \leq n$), where $\mu \geq 1$. It means $r \leq |\Omega|/(Cn \log n)$ ((2)).
Our method FGSR requires $d \geq r$. Thus, according to inequalities (1) or (2), we set $d = |\Omega|/(m+n)$.

We added truncated nuclear norm [ex1], weighted nuclear norm [ex2], and Riemannian pursuit [ex7] to the experiments.
Figure 1(a) shows that the recovery errors of the three supplemented methods are higher than those of our FGSR
methods when the missing rate is high. Note that in truncated nuclear norm, we have used the true rank (though
difficult to know beforehand in practice); otherwise, the recovery error will be much higher. Figure 1(d) shows that our
FGSR methods are much faster than all methods except Riemannian pursuit. In Figure 2(a)(b), FGSR methods also
outperformed Riemannian pursuit. In the noisy cases (Figure 2), FGSR was solved by PALM (faster than ADMM used
in the noiseless case) and its time costs are within $[1s, 3s]$, while Riemannian pursuit's time costs are within $[1.5s, 2.5s]$.
Note that the code of Riemannian pursuit was written by mixed programming C&MATLAB, which is much faster than
pure MATLAB (utilized in all other methods). In Figure 3, FGSR methods outperformed Riemannian pursuit on real
data. In sum, our FGSR methods are more accurate than all other methods. In terms of computational cost, FGSR
methods are comparable to Riemannian pursuit and are much faster than other methods.

Figure 1: Matrix completion on noiseless synthetic data: (a) different missing; (b)(c) different rank initialization (missing rate = 0.6 or 0.7); (d) computational cost (missing rate = 0.7).

**Response to reviewer** #2    We added the results of FGSR-1/2 in the experiments (shown in Figures 1, 2, and 3).
FGSR-1/2 is more accurate than FGSR-2/3. We also added the comparison of the improved case of Bi-nuclear norm
(S-2/3, Shang et al. TPAMI2017), which is denoted by $F^2$+Nuclear norm. As shown in Figure 1, our FGSR-1/2 and
FGSR-2/3 are slightly more accurate and much faster than Bi-nuclear norm (S-1/2) and $F^2$+Nuclear norm (S-2/3).
Similar comparative results can be found in the noisy cases and the results of Bi-nuclear norm, $F^2$+Nuclear norm,
Schatten-2/3, and Schatten-1/4 were omitted in Figure 2 for simplicity.

Figure 2: Matrix completion on noisy synthetic data: (a)(b) recovery error when SNR = 10 or 5; (c) the effect of rank initialization (SNR = 10, missing rate = 0.5); (d) the effect of $p$'s value in Schatten-$p$ norm (solved by FGSR if $p < 1$).

**Response to reviewer** #3    Our motivation is to provide a class of SVD-free and accurate nonconvex regularizations
for matrix rank with theoretical guarantees. We improved the illustration of our motivation according to your suggestion.
The numerical results (e.g. Figure 2(d)) showed that smaller $p$
leads to lower recovery error but the improvement is not significant
when $p$ is too small (e.g. <2/5). The phenomenon is consistent with
our generalization error bound. Therefore, in practice, we suggest
using $p = 2/3$ or $1/2$ because they are faster than $p \leq 2/5$. The
results of FGSR-1/2 have been added to Figures 1, 2, and 3. FGSR-
1/2 is more accurate than FGSR-2/3 but is slightly slower. In sum,
we suggest FGSR-1/2 if time cost is relatively less demanding.

Figure 3: NMAE on Movielens-1M data

[Meta-Review · NeurIPS 2019]

Schatten norms are often useful for convex relaxations for matrix factorisation problems. This paper proposes a so called Factor Group-Sparse Regularizers (FGSRs), that give an alternative formulation For certain Schatten-p norms, where p is small. These relaxations are claimed having useful properties: they are tighter surrogates for the rank than the nuclear norm and they can be optimized without an explicit SVD calculation (which can become quite prohibitive even if randomized methods are used). The authors also provide a computational study to demonstrate the effectiveness of the approach and illustrate that the method seem to perform well in in denoising and matrix completion tasks regardless of the initial choice of rank. The paper has a fine balance in theoretical and simulation study, and makes some interesting observations about FGSRs. Overall, the study is well executed and well reported. However, the topic is a well investigated one, so inevitably many interesting improvements become somewhat incremental, including the contributions of this paper.